# Sustainable production of dopamine hydrochloride from softwood lignin

Lin Dong[1,2], Yanqin Wang [2]✉, Yuguo Dong[1], Yin Zhang[1], Mingzhu Pan [1], Xiaohui Liu [2], Xiaoli Gu [1], Markus Antonietti [3] & Zupeng Chen [1,4]✉

Dopamine is not only a widely used commodity pharmaceutical for treating neurological diseases but also a highly attractive base for advanced carbon materials. Lignin, the waste from the lignocellulosic biomass industry, is the richest source of renewable aromatics on earth. Efficient production of dopamine direct from lignin is a highly desirable target but extremely challenging. Here, we report an innovative strategy for the sustainable production of dopamine hydrochloride from softwood lignin with a mass yield of 6.4 wt.%. Significantly, the solid dopamine hydrochloride is obtained by a simple filtration process in purity of 98.0%, which avoids the tedious separation and purification steps. The approach begins with the acid-catalyzed depolymerization, followed by deprotection, hydrogen-borrowing amination, and hydrolysis of methoxy group, transforming lignin into dopamine hydrochloride. The technical economic analysis predicts that this process is an economically competitive production process. This study fulfills the unexplored potential of dopamine hydrochloride synthesis from lignin.

Dopamine is not only a commodity pharmaceutical for regulating renal function, blood pressure, and neurobehavioral disorders (e.g., Alzheimer's disease, Parkinson's disease, schizophrenia, mania, depression, substance abuse, and eating disorders)[1–5] but is also widely used in the field of polydopamine-based materials due to its biocompatibility and self-polymerization capability (Supplementary Fig. 1)[6,7]. Particularly, dopamine-based N-doped carbon is the most important asset of future carbon electrodes, as the oxidative polymerized polydopamine is a very strong ion and electron conductor[8–11]. The global dopamine market revenue was more than 320 million USD in 2022, and the global dopamine demand is anticipated to grow at a rate of 8.2% annually in the next decade[12]. Thus, dopamine has attracted considerable interest all over the world in the past few years[13–15]. Dopamine hydrochloride is predominantly produced from

vanillin via a sequence of condensation, reduction, and hydrolysis (Fig. 1a)[16]. However, vanillin and the reaction intermediates have poor stability and are inclined to produce high boiling point by-products, which can hardly be separated by conventional distillation techniques. Moreover, the consumption of large amounts of Zn and Hg in the reduction step causes serious environmental restrictions. Besides chemical synthesis processes, an enzymatic system was also developed for the production of dopamine from catechol (Fig. 1b)[17]. 3,4-dihydroxy-L-phenylalanine (L-DOPA) is first synthesized from catechol, pyruvate, and ammonia over a biocatalyst, followed by enzymatic conversion of L-DOPA to dopamine by decarboxylase. However, this route largely depends on fossil resources and has several disadvantages, such as long reaction time, tedious separations, and high costs. Therefore, a more sustainable and economical strategy for the

[1]Jiangsu Co-Innovation Center of Efficient Processing and Utilization of Forest Resources, International Innovation Center for Forest Chemicals and Materials, College of Chemical Engineering, Nanjing Forestry University, Longpan Road 159, 210037 Nanjing, China. [2]Key Laboratory for Advanced Materials and Joint International Research Laboratory of Precision Chemistry and Molecular Engineering, Feringa Nobel Prize Scientist Joint Research Center, Research Institute of Industrial Catalysis, School of Chemistry and Molecular Engineering, East China University of Science and Technology, 200237 Shanghai, China. [3]Department of Colloid Chemistry, Max-Planck Institute of Colloids and Interfaces, Research Campus Golm, Am Mühlenberg 1, Potsdam 14476, Germany. [4]Leibniz-Institute for Catalysis, University of Rostock, Albert Einstein Street, 29a, Rostock 18059, Germany. ✉e-mail: wangyanqin@ecust.edu.cn; czp@njfu.edu.cn

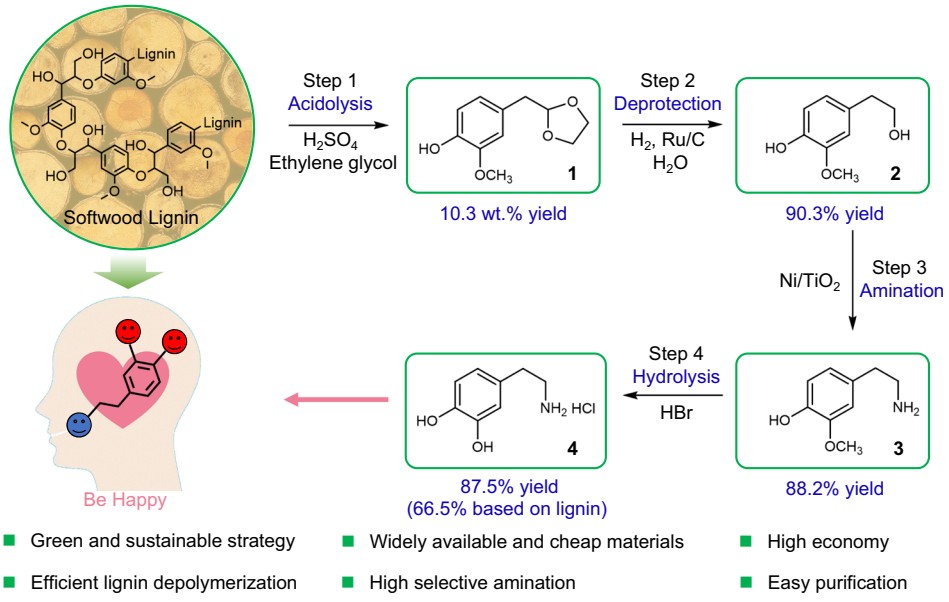

**a** Traditional route for dopamine hydrochloride production from vanillin

- Poor stability
- Difficult separation
- Pollution

**b** Traditional route for dopamine production from catechol

- Fossil resource
- High energy consumption
- Harsh condition

**c** Sustainable route for dopamine hydrochloride production from softwood lignin (This work)

- Green and sustainable strategy
- Widely available and cheap materials
- High economy
- Efficient lignin depolymerization
- High selective amination
- Easy purification

**Fig. 1 | Strategies employed for dopamine production. a** Traditional route for dopamine hydrochloride production from vanillin. **b** Traditional route for dopamine production from catechol. **c** Lignin-to-dopamine hydrochloride route developed in this work. The yield of each step was indicated underneath the targeted products.

efficient production of dopamine from renewable resources is still highly desired and would have great industrial and social significance[18,19].

Lignin is the only abundant and renewable aromatic source on Earth, accounting for 15–35% by weight of lignocellulosic biomass that is harvested from forestry and agricultural activity worldwide, with an annual production of isolated lignin of ca. 50 million metric tons[20–22]. Unfortunately, lignin has always been treated as a waste product and used as low-value fuel in the pulp and paper industry, where it is burned to supply energy and to recover pulping chemicals in the operation of paper mills[23]. Recently, some literature described the conversion of lignin-derived platform chemicals to amines (Supplementary Fig. 2)[24–27], but the reactants are limited to lignin model compounds rather than natural lignin, as well as complex product distribution, separation difficulty, low value-added, and harsh reaction conditions. With careful examination of the framework of lignin, the dominant guaiacyl (G) units can set also the base of the structure of dopamine, which provides considerable potential for its production from the abundant biopolymer lignin. Hence, using lignin as a sustainable and desirable candidate feedstock to produce dopamine

hydrochloride is a tantalizing strategy to introduce sustainable chemistry to generate a high-value compound.

Some breakthrough approaches have been developed for the sustainable production of high-value pharmaceuticals from lignin or lignin derivatives. The main products obtained from lignin depolymerization include guaiacyl, syringyl, and tricin derivatives, which can be converted into chemicals with pharmacological effects, including balsacones, menthol, carbazoles, menthol, and so on[28,29]. For example, Jess et al. reported an effective synthetic strategy to produce menthol from lignin derivatives via the preparation of thymol and its hydrogenation[30]. Barta et al. presented a cleave and couple process to selectively produce antidepressants, by combining the lignin depolymerization and the formation of C–C/C–N bonds[31,33]. However, the resulting products are always mixtures with high boiling points, which demand complicated and energy-consuming processes for their purification and separation.

In addition, the low yield of target products complicates operations on an industrial scale. Therefore, the highly effective conversion of lignin into an easily purifiable, single-component, value-added pharmaceutical still remains a challenge.

Herein, we propose an innovative lignin-to-dopamine hydrochloride strategy, which results in the sustainable production of highly pure dopamine that can be simply separated by filtration with a mass yield of 6.4 wt% based on lignin (a yield of 66.5% on theoretical amounts of monolignols). As shown in Fig. 1c, softwood lignin was first acid-catalyzed depolymerized into 2-(4-hydroxy-3-methoxyphenyl)−1,3-dioxolane (**1**) with ethylene glycol as a stabilization agent (step 1). Then, the dioxolane group in the **1** was removed by hydrolysis and hydrogenation, resulting in the formation of 2-(4-hydroxy-3-methoxyphenyl)-ethanol (**2**) (step 2). Next, hydrogen-borrowing amination of the hydroxy group afforded the 2-(4-hydroxy-3-methoxyphenyl)-ethamine (**3**) over nickel-based catalysts (step 3). Finally, **3** was converted into dopamine hydrobromide by the hydrolysis of the methoxy group, followed by further conversion into dopamine hydrochloride (**4**) in ethanol (step 4). The dopamine hydrochloride was obtained in the form of a white powder that appeared at the bottom, thus facilitating its purification from the reaction mixture.

## Results

### Depolymerization of lignin into monomers

Lignin is mainly an amorphous tridimensional polymer of three monolignols, known as syringyl (S), guaiacyl (G), and *p*-hydroxyphenyl (H) units. These monolignols differ in the number of methoxy groups, namely, the S unit has two methoxy groups, the G unit has one methoxy group, and the H unit has none[34]. As the dopamine hydrochloride molecular structure contains two adjacent hydroxy groups in the aromatic moiety, guaiacyl is the most suitable unit for the synthesis of dopamine hydrochloride without removing or adding groups attached to the aromatic moiety. According to previous reports[20], the content of each monolignol in lignin is related to plant taxonomy. Softwood lignin contains more G units with a proportion of 75–90%, while hardwood lignin and grass lignin contain much fewer G units (only 25–50%). Therefore, softwood is selected to obtain lignin in this work, including spruce, pine, cedar, and Douglas fir. The relative ratio of the main linkages and subunits of lignin was determined by 2D heteronuclear single-quantum correlation-nuclear magnetic resonance (2D-HSQC NMR), which is widely used as a standard protocol for analyzing lignin, and the results were summarized in Supplementary Figs. 3, 4 and Supplementary Table 1. Relative quantification of the main linkages in spruce lignin provided a $\beta$-O-4/$\beta$–5/$\beta$-$\beta$ ratio of 0.68/0.28/0.04 and H/G/S subunit ratio of 0/0.98/0.02. Because of the high ratio of $\beta$-O-4 linkages and G subunit, spruce lignin was used to identify the effective catalysts and optimize the reaction conditions.

Despite efficient strategies for lignin depolymerization that have been developed over the past few decades (e.g., acid-catalyzed, base-catalyzed, reductive, oxidative, pyrolytic, and photocatalytic depolymerization)[21,22,35], they normally lead to the formation of C3-fragments, namely, the side chain of products contains three carbons (e.g., propane, propanol). For example, when the reductive depolymerization of spruce lignin was conducted over Pd/C under a hydrogen atmosphere, the major product was C3-fragment 2-(4-hydroxy-3-methoxyphenyl)-propanol (**1a**) (Supplementary Fig. 5a). However, the side chain of dopamine hydrochloride contains only two carbons (ethamine), therefore preferring C2-fragments as the precursor. Different from reductive depolymerization, acid-catalyzed depolymerization involves the loss of formaldehyde, leading to the formation of C2-fragments (Supplementary Fig. 6). Unfortunately, no C2-fragment, namely 2-(4-hydroxy-3-methoxyphenyl)-acetaldehyde (**1b**), was detected with $H_2SO_4$ as an acid catalyst in 1,4-dioxane solvent, which could be attributed to the condensation of this unstable **1b** (Supplementary Fig. 5b). To this end, Barta et al. proposed an in situ stabilization strategy via capturing unstable compounds by reaction with diols[36–39], which inspired us to convert lignin into C2-fragments via aldolization with aldehydes in the presence of an acid, using ethylene glycol (EG) as the stabilization agent (Supplementary Fig. 7).

Theoretical monomer yield of lignin acid-catalyzed depolymerization is dependent on the $\beta$-O-4 linkages content of the lignin, which was calculated by combining the 2D-HSQC NMR (Supplementary Figs. 3, 4) and derivatization followed by reductive cleavage (DFRC) methods (Supplementary Fig. 8). As shown in Table 1, the theoretical yield of **1** in spruce lignin, pine lignin, cedar lignin, and douglas fir lignin was 10.8, 8.8, 5.4, and 9.5 wt%, respectively. The solvents and acid catalysts were screened, which were found to impose significant influences on the yield of **1** (Supplementary Table 2). Among the investigated solvents and acid catalysts, 1,4-dioxane and $H_2SO_4$ showed the highest performance and the yield of **1** was up to 10.3 wt% with depolymerization efficiency of 95.4%. Other by-products in lignin oil included 2-(4-hydroxyphenyl)−1,3-dioxolane, other monomers, and dimers, whose structure and mass yields were summarized in Supplementary Table 3 and Supplementary Fig. 9. A molecular weight average typical for raw spruce lignin ($M_n = 1947$ g mol$^{-1}$ and $M_w = 4522$ g mol$^{-1}$) was confirmed by gel permeation chromatography (GPC) analysis (Supplementary Fig. 10). After lignin depolymerization, the molecular weight average decreased dramatically ($M_n = 1048$ g mol$^{-1}$ and $M_w = 1789$ g mol$^{-1}$), illustrating that the linkages between subunits in lignin were broken efficiently. The 2D-HSQC NMR spectra of spruce lignin before and after depolymerization were shown in Supplementary Figs. 3, 11, confirming the structures of the three major linkages ($\beta$-O-4, $\beta$–5, and $\beta$–$\beta$) were effectively changed due to the cleavage of C−O bonds and the structures with 1,3-dioxolane end groups were formed. Next, pine, cedar, and Douglas fir lignin were employed as feedstocks. The yield of **1** for conversion of pine, cedar, and Douglas fir lignin was determined to be 8.6, 5.2, and 7.9 wt%, respectively, demonstrating the universality of the system.

### Conversion of 1 into 2

The dioxolane group of **1** is required to be translated into a hydroxyl or aldehyde group, providing the possibility for further amination. The highest **1** yield was obtained in 1,4-dioxane solvent over $H_2SO_4$ catalyst, thus the deprotection of **1** was also preferentially conducted in 1,4-dioxane in the presence of $H_2SO_4$, resulting in the integration of steps 1 and 2 in one-pot. Three deprotection strategies were proposed, including hydrogenolysis, hydrolysis, and combined hydrolysis and hydrogenation (Fig. 2). The hydrogenolysis of **1** was conducted over Ru/C catalyst in 1,4-dioxane at 120 °C and an initial $H_2/N_2$ (50 vol%) pressure of 1.0 MPa for 1 h, yielding **1b** (23.3%) and **2** (0.7%) at a conversion of 43.1% (Fig. 2a). Meanwhile, a certain amount of 4-ethyl-2-methoxyphenol (**2a**) (11.5%) was produced as a by-product. In the hydrogenolysis route, **1** was first converted into **1b**, followed by hydrogenation to produce **2**. However, **2** could be easily hydrodeoxygenated resulting in the formation of **2a** by-product, which excludes the possibility of amination in the next step. In addition, the hydrolysis of **1** was also conducted by adding some amount of $H_2O$ (Fig. 2b), which exhibited improved conversion (80.9%) and yield of **1b** (77.5%). However, the yield of **1b** decreased significantly from 77.5% to 7.3% with prolonged reaction time (Supplementary Table 4, entry 3), which was caused by the condensation of **1b** in the presence of acid. Delightedly, when both Ru/C catalyst and $H_2O$ exist in the reaction system under a hydrogen atmosphere, the yield of **2** could be increased up to 90.3% with near-complete conversion (99.7%) (Fig. 2c). It is worth noting that the **2** was the only liquid product, which leads to easy separation with conventional technology. No obvious decrease in **2** formation with prolonged reaction time (Supplementary Table 4, entry 5), suggesting that the **2** was stable in this reaction system. In this process, the **1** was first converted into **1b** via hydrolysis, which was further hydrogenated to produce **2** using hydrogen as the stabilization agent (Supplementary Fig. 12).

### Hydrogen-borrowing amination of 2 into 3

Ni- and Ru-based catalysts have demonstrated high selectivity in the synthesis of primary amines from alcohols via hydrogen-borrowing

**Table 1 | Acid-catalytic lignin depolymerization with ethylene glycol stabilization[a]**

Softwood lignin

| Entry | Lignin | Yield of 1 (wt%) | Theoretical yield of 1 (wt%)[b] | Depolymerization efficiency (%)[c] |
|---|---|---|---|---|
| 1 | Spruce | 10.3 | 10.8 | 95.4 |
| 2 | Pine | 8.6 | 8.8 | 97.7 |
| 3 | Cedar | 5.2 | 5.4 | 96.3 |
| 4 | Douglas fir | 7.9 | 9.5 | 83.2 |

[a]Reaction conditions: lignin (0.2 g), ethylene glycol (0.72 mL), 1,4-dioxane (30 mL), $H_2SO_4$ (16 µL), temperature (140 °C), 1 h.
[b]Theoretical yield of 1 was obtained by combining the 2D-HSQC NMR and DFRC methods. 1 represents 2-(4-hydroxy-3-methoxyphenyl)-1,3-dioxolane.
[c]Depolymerization efficiency (%) = yield of 1 / theoretical yield of 1.

amination mechanism[40–43]. Therefore, the amination of **2** into **3** was conducted in a fix-bed reactor using different Ni- and Ru-based catalysts (i.e., 10%Ni/$TiO_2$, 10%Ni/$Al_2O_3$, 65%Ni/$Al_2O_3$, Raney Ni, 5%Ru/$CeO_2$, and 5%Ru/C) (Fig. 3a). The reaction temperature was first screened over different catalysts to identify the optimal temperature (Supplementary Fig. 13). It is obvious that 10%Ni/$TiO_2$ shows the unique activity under the investigated conditions, with 100% conversion and **3** yield up to 88.2% at 160 °C. Meanwhile, the yield of **1b** and dimer was 6.9% and 0.7%, respectively. With the reaction temperature increasing from 160 to 200 °C, the overall conversion was unaltered but the yield of **3** decreased slightly from 88.2 to 74.8% (Supplementary Fig. 13a), which was attributed to the formation of dimer at high temperature. Under the optimized reaction conditions, other Ni-based catalysts (i.e., 10%Ni/$Al_2O_3$, 65%Ni/$Al_2O_3$, and Raney Ni) showed inferior performance. For example, the yield of **3** and conversion over 10%Ni/$Al_2O_3$ was only 59.7% and 87.0%, respectively. When increasing the loading of Ni to 65%, the yield of **3** could be increased to 86.0%, which is unsatisfactory considering the relatively high metal content. On the other hand, the performance of 10%Ni/$Al_2O_3$ catalysts was highly sensitive to reaction temperature (Supplementary Fig. 13b, c), leading to a narrow range of optimal conditions. Meanwhile, the Raney Ni catalyst showed a much poorer yield of **3** formation (52.8%). In the cases of Ru-based catalysts, 5%Ru/$CeO_2$ produced **3** with a yield of 78.0% at 100% conversion. Similar results were obtained over 5%Ru/C. The constant conversion of **2** and yield of **3** formations were identified in a continuous 20 h test on stream, highlighting the stability of the optimal 10%Ni/$TiO_2$ catalyst and excluding the effects of deactivation caused by leaching or aggregation of nickel (Fig. 3b). Furthermore, the post-analysis of the used catalyst by transmission electron microscopy (TEM), X-ray powder diffraction (XRD), inductively coupled plasma optical emission spectroscopy (ICP-OES), and nitrogen physisorption confirms the virtually identical structure, physical properties, and metal content to the fresh one (Supplementary Figs. 14, 15, and Supplementary Table 5). According to previous reports[44–46], the reaction network of hydrogen-borrowing amination of **2** was proposed (Supplementary Fig. 16). At the very beginning of the reaction, **2** is dehydrogenated into **1b** and a stoichiometric $H_2$ molecule, which is believed to be the rate-determining step in the overall reaction and needs the participation of metal catalysts[47]. Then the amination between **1b** and $NH_3$ occurs spontaneously (Supplementary Fig. 16)[47], which is hydrogenated into **3** with the self-supplied $H_2$ that is produced in the first step. Therefore, the overall process does not require extra hydrogen.

## Hydrolysis of 3 into 4

The acid system is reported to be one of the most active and selective catalysts for the demethylation of aromatic methyl ethers, such as the hydrolysis of guaiacol into catechol[48–50], and therefore, different acids were utilized to convert **3** into dopamine (Fig. 4a). The hydrolysis of **3** was first conducted at 120 °C for 6 h with 1.0 MPa $N_2$ in the presence of different acids. The time profiles of dopamine production suggest that the amount of dopamine increased gradually prolonging the reaction time, without the formation of other intermediates or by-products (Fig. 4b). Significantly, the reaction over HBr reached a yield of 94.5% for dopamine hydrobromide (**4a**) with the almost full conversion of **3**. In addition, nearly no other product was detected in high-performance liquid chromatography (HPLC), suggesting high purity of **4a** (Supplementary Fig. 17). More importantly, **4a** could be obtained in the form of white powder that appeared at the bottom of the reactor after the reaction, which largely facilitates its purification from the mixture simply by filtration and drying. In addition, the HBr can be easily reused for consecutive tests, which shows similar hydrolysis activity (Supplementary Fig. 18). For the HI catalyst, the yield of dopamine hydroiodate (**4b**) was only 50.1% with partial **3** remaining, showing lower hydrolysis activity than that of HBr. In addition, no solid **4** was precipitated out over the HCl catalyst as **4** was soluble in HCl. Next, the

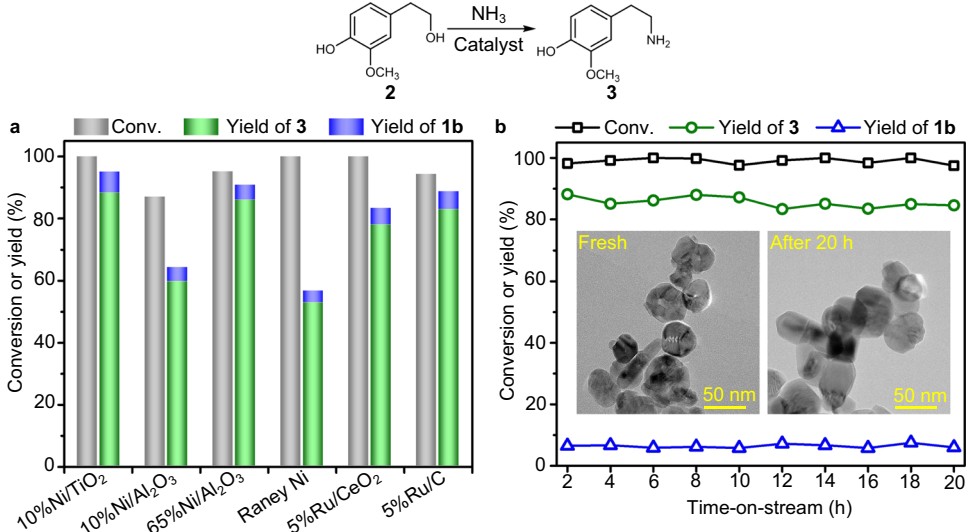

**Fig. 2 | Establishing optimal reaction system for the deprotection of 1 to 2.** Catalytic results for the conversion of **1** with various systems in a batch reactor. Reaction conditions: substrate (0.02 g), 1,4-dioxane (30 mL), $H_2SO_4$ (16 µl), temperature (120 °C), 1 h. **a** Ru/C (0.1 g), $H_2/N_2$ (50 vol%, 1.0 MPa). **b** $H_2O$ (1 mL), $N_2$ (1.0 MPa). **c** Ru/C (0.1 g), $H_2/N_2$ (50 vol%, 1.0 MPa), $H_2O$ (1 mL).

**Fig. 3 | Catalytic hydrogen-borrowing amination of 2 to 3 and stability test.** **a** Catalytic result for the amination of a **2** in a fix-bed reactor over different catalysts. Reaction condition: catalyst (0.5 g), **2** in *p*-xylene solution (0.025 M), $NH_3$ pressure (0.6 MPa), liquid flow rate (0.3 mL min⁻¹), gas flow rate (100 mL min⁻¹), WHSV (0.163 h⁻¹). The reaction temperature of 10%Ni/TiO₂, 10%Ni/Al₂O₃, 65%Ni/Al₂O₃, Raney Ni, 5%Ru/CeO₂, and 5%Ru/C was 160, 180, 170, 160, 160, and 140 °C, respectively. **b** Stability test of 10%Ni/TiO₂ catalyst for the amination of **2**–**3** via hydrogen-borrowing amination. Insets represent the corresponding TEM images for the fresh and used catalysts. Reaction conditions: 10%Ni/TiO₂ (0.5 g), **2** in *p*-xylene solution (0.025 M), temperature (160 °C), $NH_3$ pressure (0.6 MPa), liquid flow rate (0.3 mL min⁻¹), gas flow rate (100 mL min⁻¹).

conversion of **4a** into **4** was conducted with the presence of HCl in ethanol (Fig. 4c). After the reaction, a large amount of the targeted **4** could be obtained in the form of white powder precipitates at the bottom of the vial. The white powder could be easily separated by a simple filtration and drying process and was further analyzed by a series of methods, confirming that the yield and purity of **4** were 92.6% and 98.0%, respectively (Supplementary Fig. 19). ¹H and ¹³C NMR spectra for the obtained white powder were shown in Fig. 4d, which were consistent with the results of the previous studies[51]. Considering the long-standing challenge of efficient isolation and purification of products from the reaction mixture in biorefinery, the spontaneous precipitation of desired dopamine from solution is impressive for the practical application of biomass valorization, which can reduce the cost of post-treatment significantly.

## Catalytic conversion of lignin to 4

In the conversion of spruce lignin into **4**, the mass yields and molar yields based on lignin, and the yield of each step were summarized in Fig. 5a. After acid-catalyzed depolymerization of spruce lignin, the mass yield of **1** was 10.3 wt% with a depolymerization efficiency of 95.4%. After the deprotection and amination reaction, the yield of **3** was 6.5 wt%. Finally, 6.4 wt% (341 µmol/g) of **4** was obtained based on spruce lignin. A tenfold scaling-up experiment with sole spruce lignin as substrate was carried out, in which the details of the process of **4** synthesis, mass yields of products, product purification, solvent recovery, catalyst, and additive separation were shown in Supplementary Fig. 20. Though the yield of **4** was only 3.3 wt% (94.7% purity), the high value-added and huge market of **4** were attractive. On the basis of our experimental results, a process model to perform a

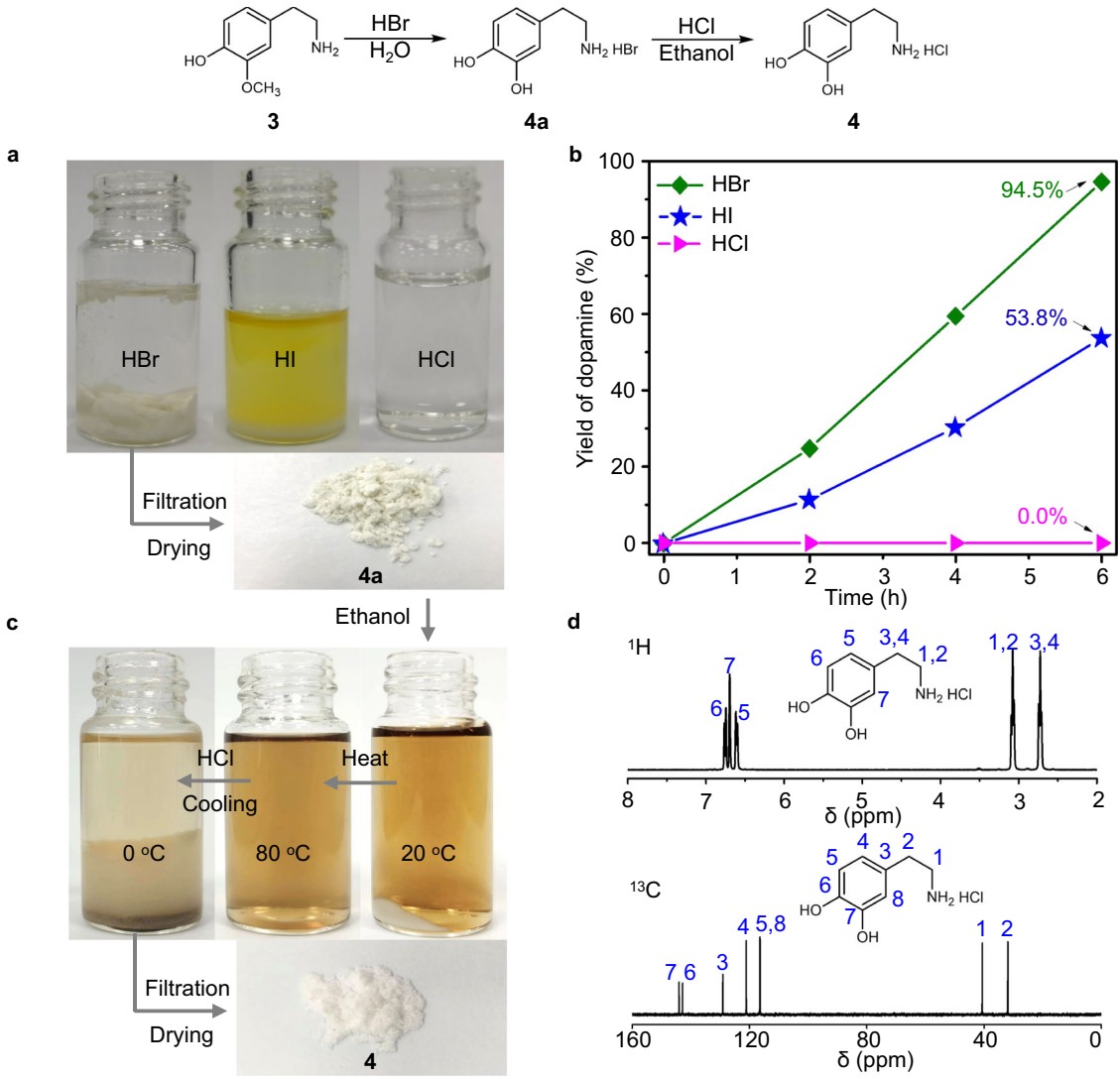

**Fig. 4 | Catalytic hydrolysis of 3 to 4 and product separation. a** Digital photo of the reaction mixtures after the hydrolysis of **3** into **4a** in an aqueous solution over different acids. **b** Catalytic results for the hydrolysis of a **3** in an aqueous solution over different acids. Reaction condition: **3** (1.0 g), acid (6.0 g), $N_2$ pressure (0.1 MPa), temperature (120 °C). The concentrations of HBr, HI, and HCl were 40.0, 55.0–58.0, and 36.0–38.0 wt%, respectively. **c** Digital photo of the reaction mixtures after the conversion of **4a** into **4** in ethanol. **d** $^1H$ and $^{13}C$ NMR spectra for the obtained **4** from (**c**). NMR spectra were recorded on Bruker Avance-400 instrument, using $D_2O$ as solvent.

techno-economic analysis was conducted to evaluate the economic potential (Fig. 5b, c). The production cost of dopamine hydrochloride in this work was 2.20 million CNY/t (Chinese Yuan per ton) (Supplementary Tables 6–9), which was significantly lower than the dopamine hydrochloride market prices in the range of 4–6 million CNY/t over the last year[12]. These results indicate that the lignin-to-dopamine hydrochloride route is an economically competitive process, which imposes great potential to be translated into the lignocellulosic biorefinery.

## Discussion
We have presented an efficient sustainable route for producing dopamine hydrochloride direct from renewable softwood lignin, achieving dopamine hydrochloride with a purity of 98.0% obtained simply via filtration and drying. An overall mass yield of 6.4 wt% can be achieved from softwood lignin over an extraction-four step derivatization cascade if concerning the theoretical lignin content. This work provides an example of the transformation of non-edible renewable biomass waste into high-value dopamine hydrochloride that has important applications in pharmaceutical manufacturing and future

N-doped carbon electrodes, which significantly enhances the overall economic feasibility of a lignocellulosic biorefinery.

## Methods
### Lignin extraction method
Spruce lignin was extracted according to a literature procedure[52] and details of the experiments were as follows. In a round-bottom flask (1 L) with a condenser, pre-ground birch wood (20 g) and methanol (200 mL) containing 3 wt% HCl were combined. The mixture was refluxed for 12 h under stirring and cooled to room temperature. The residue was removed by filtration and washed with methanol. The filtrate was concentrated to 50 ml by rotary evaporation and then poured into ice-cold water with vigorous stirring, resulting in a brown solid precipitate. This lignin was collected by filtration, washed, and dried overnight. Pine lignin, cedar lignin, and Douglas fir lignin were extracted in the same way. The lignin yield of spruce, pine, cedar, and Douglas fir was 18.6, 20.1, 17.3, and 16.7 wt%, respectively, which was nearly the lignin content in softwood (25–30 wt%)[20].

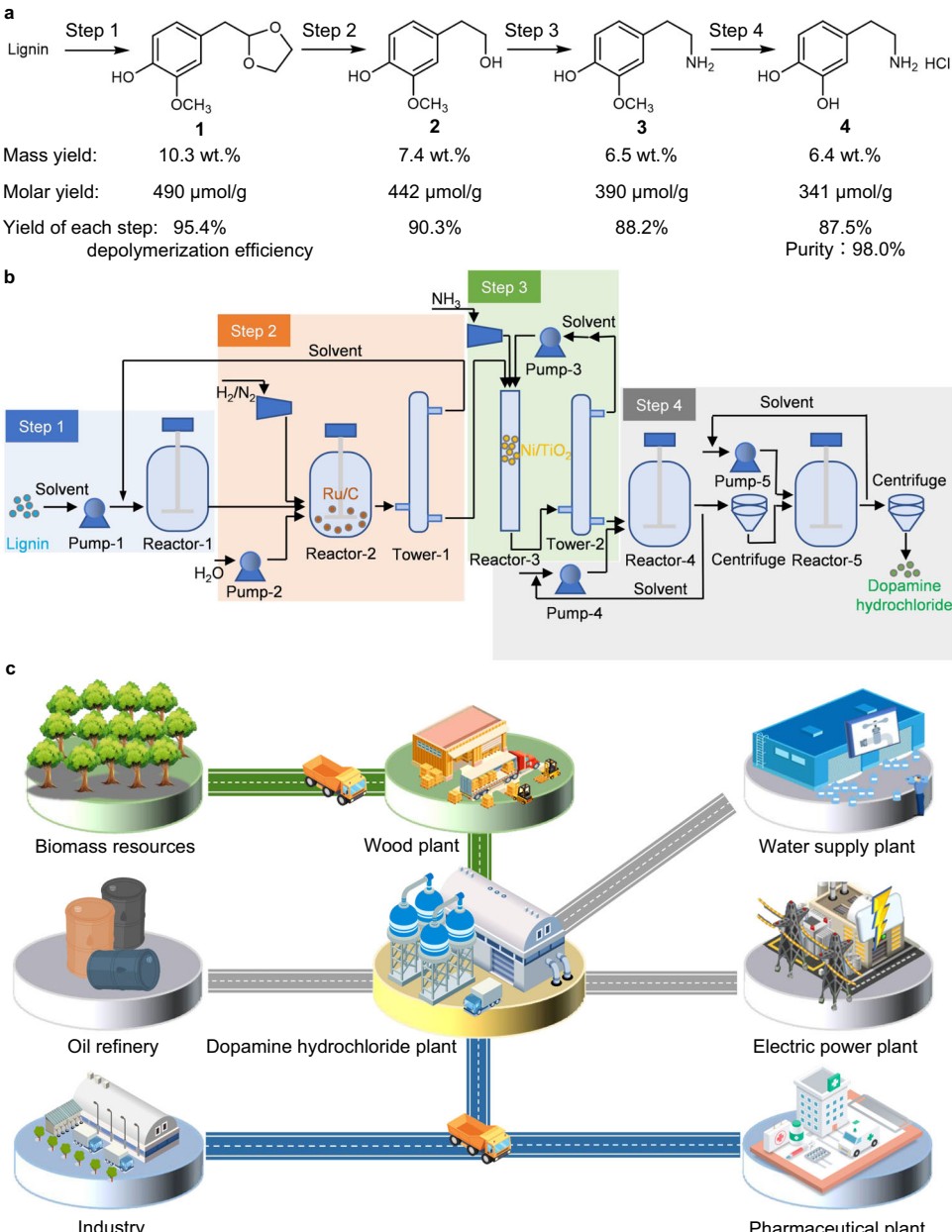

**Fig. 5 | Catalytic conversion of lignin to dopamine hydrochloride. a** The yields of the products in the spruce lignin-to-dopamine hydrochloride route. Mass yield = ( mass of product)/(mass of lignin)×100%; Molar yield = (molar of product)/(mass of lignin); Yield of each step = (molar of product)/(molar of the substrate in each step)×100%; Depolymerization efficiency = (yield of **1**/theoretical yield of **1**)×100%; theoretical yield of **1** in lignin was obtained by combining the 2D-HSQC NMR method and derivatization followed by the reductive cleavage method. **b** Conceptual process modeling for dopamine hydrochloride production from lignin. The process model integrates the four catalytic steps: (i) acid-catalyzed lignin depolymerization; (ii) deprotection reaction of **1**; (iii) hydrogen-borrowing amination of **2**; and (iv) hydrolysis reaction of **3** into **4**. **c** Conceptual diagram for dopamine hydrochloride production from lignin. It is comprised of the production and application of dopamine hydrochloride.

## Catalyst preparation

The Ru-based catalysts (5%Ru/C and 5%Ru/CeO$_2$) catalysts were prepared by the incipient wetness impregnation method with appropriate amounts of an aqueous solution of RuCl$_3$ 3H$_2$O. The obtained sample was dried at 100 °C for 12 h and then reduced in a 10% H$_2$/Ar flow at 400 °C for 4 h. The ruthenium loading in each catalyst was at 5 wt%.

The Ni-based catalysts (10%Ni/TiO$_2$, 10%Ni/Al$_2$O$_3$, and 65%Ni/Al$_2$O$_3$) catalysts were also prepared by the incipient wetness impregnation method with appropriate amounts of an aqueous solution of Ni(NO$_3$)$_2$ 6H$_2$O. The obtained sample was dried at 100 °C for 12 h and then baked at 500 °C for 4 h, followed by reduced in a 10% H$_2$/Ar flow at 400 °C for 4 h. The nickel loading in each catalyst was at 10 or 65 wt%.

## Catalyst activity tests

The detailed reaction conditions are described in the figure captions and table footnotes.

*Step 1*: The acid-catalyzed depolymerization of lignin was carried out in a 50 mL stainless-steel autoclave. In a typical reaction, lignin (0.2 g), acid (16 μl), ethylene glycol (0.72 mL), and solvent (30 mL) were filled into the reactor, which was then sealed, purged three times with N$_2$, and charged to an initial pressure of 0.1 MPa with N$_2$. The reactor was heated to 140 °C, and the reaction was conducted with magnetic stirring for 1 h. After the reaction, the reactor was quenched in an ice-water bath. The organic phase was qualitatively analyzed by GC−MS (Shimadzu QP2020 NX) and quantitatively analyzed by GC-FID

(Agilent 7890B) with an HP-5 column (30 m × 250 μm). The column temperature began at 50 °C (held for 5 min) and was then raised at 10 °C min$^{-1}$ to 280 °C (held for 1 min). Pentadecane was used as an internal standard for the quantification of liquid products.

*Step 2*: The deprotection of **1** was also carried out in a 50 mL stainless-steel autoclave. In a typical reaction, **1** (0.02 g), $H_2SO_4$ (16 μl), $HO_2$ (1 mL), and 1,4-dioxane (30 mL) were filled into the reactor, which was then sealed, purged three times with $H_2/N_2$ (50 vol%), and charged to an initial pressure of 1.0 MPa. Then, the autoclave was heated to 120 °C for 1 h. After the reaction, the reactor was quenched in an ice-water bath immediately, and the reaction mixture was centrifuged to separate the catalyst. The qualitative and quantitative analysis methods of products were the same as above in Step 1.

*Step 3*: The hydrogen-borrowing amination of **2** was tested in a fixed-bed reactor system. Before the test, 0.5 g of catalyst packed into the middle portion of the stainless-steel tubular reactor (inner diameter of 5 mm, length of 30 cm) was in situ activated at 400 °C for 4 h with $H_2/Ar$ under atmospheric pressure. Then a feed of **2** in *p*-xylene solution (0.025 M) was injected into the reactor by an HPLC pump, which resulted in a weight-hourly space velocity (WHSV) of 0.163 h$^{-1}$. The amination of **2** was conducted at 160 °C and 0.6 MPa $NH_3$ with the $NH_3$ flow rate of 100 mL/min. The liquid phase was separated from the gas phase and collected by a gas–liquid separator. The liquid phase analysis was the same as above in Step 1.

*Step 4*: The hydrolysis of **3** into **4a** was carried out in a 10 mL vial. In a typical reaction, the **3** (1.0 g) was mixed with an acid (6 mL) and reacted at 120 °C for 6 h in an oil bath under an $N_2$ atmosphere. After the reaction, the reactor was quenched in an ice-water bath, and large amounts of **4a** in the form of white powder precipitated at the bottom. After the hydrolysis reaction, white powder was obtained by filtration and drying and was further dissolved in water. The aqueous phase was qualitatively analyzed by HPLC (Agilent 1200 Series) equipped with an Agilent C18 column (Zorbax SB-C18; 4.6 mm × 150 mm, 3.5 μm) and a diode-array detector. 40 μL of the reaction mixture was taken out in a centrifuge tube, 960 μL of water was added to the mixture, and the diluted liquid was filtered through a 0.22 μm filter. 10 μL of the sample was injected under the following conditions: column temperature = 30 °C, mobile phase = acetonitrile (20% vol/vol) with water (80% vol/vol), and flow rate = 0.5 mL min$^{-1}$, wavelength of 280 nm. The conversion of **4a** into **4** was carried out in a 10 mL vial. Firstly, **4a** (1.0 g) was mixed with ethanol (8 mL) under an $N_2$ atmosphere at 20 °C, which dissolved most of **4a**. Then, the mixture was heated to 80 °C in an oil bath and **4a** was found to be completely soluble in ethanol. Next, concentrated HCl (36.0–38.0 wt%, 1 mL) was slowly dripped into the solution. Afterward, the system was cooled to 0 °C in an ice-water bath, and a large amount of the targeted **4** could be obtained in the form of white powder precipitates at the bottom of the vial. Finally, the white powder could be easily separated by a simple filtration and drying process. The yield and purity of the white powder were determined by HPLC (Agilent 1200 Series), equipped with an Agilent C18 column (Zorbax SB-C18; 4.6 mm × 150 mm, 3.5 μm) and a diode-array detector. The mobile phase was sodium dodecyl sulfate (0.005 mol/L)/acetonitrile/glacial acetic acid/ethylenediaminetetraacetic acid disodium salt solution (0.1 mol/L) = 700/300/10/2. In brief, the obtained white powder was dissolved in the mobile phase (0.3 g/L). 10 μL of the sample solution was injected under the following conditions: column temperature = 30 °C, flow rate = 1.0 mL min$^{-1}$, and wavelength = 280 nm.

## Data availability

The authors declare that the data supporting the findings of this study are available in the article and Supplementary Information. Additional datasets related to this study are available from the corresponding authors on request.

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

## Acknowledgements

This project was supported financially by the National Natural Science Foundation of China (Nos. 22202105, 22002043, 22172048, 32171704, 22072042), the Natural Science Foundation of Jiangsu Province (BK20210608), the Natural Science Foundation of Jiangsu Higher Education Institutions of China (21KJA150003), and the China Postdoctoral Science Foundation (2023M731703). Z.C. acknowledges the support from the Alexander von Humboldt Foundation.

## Author contributions

L.D. prepared and characterized the catalysts, and performed most of the catalytic reactions. Y.D. contributed partially to the catalytic tests. Y.Z. and M.P. carried out the technical-economic assessment. X.L. contributed to 2D-HSQC NMR analysis. Y.W. and Z.C. conceived and coordinated the overall direction of the project. L.D., X. Gu., M.A., and Z.C. co-wrote the manuscript with the discussion of all co-authors.

## Competing interests

The authors declare no competing interests.
