## [Peer Review File · Nature Communications]

Reviewers' Comments:

Reviewer #1:

Remarks to the Author:

This contribution of Chen and coworkers describes the production of dopamine from lignin. This is an important molecule, and the manuscript is a valuable contribution to the array of strategies that are toward the making of amines from biomass. However i believe it is perhaps more suitable for a specialized journal such as Green Chemistry. Recently, several groups have described the formation of amines from biomass (Tao Zhang, Ning Yan - Angew Chem 2017 on primary amines from biomass, several papers from Sels , Saravanamurugan S 2022 ChemSusChem) and also reviewed by B Sels Green Chem 2017 and these rely mostly on similar catalytic systems and reductive amination or direct amination of alcohols with ammonia. Same time, also the lignin depolymerization field has been investigated extensively in the past years. This manuscript does not describe a novel method for lignin depolymerization or amination.

Some more specific comments and questions are listed below:

- 1.) The lignins have been extracted from biomass, however their yield from biomass is not given. The lignin yield from biomass after extraction can be low, so the overall obtained dopamine yield from biomass could be low. The theory monomer yield is also dependent on the b-O-4 content of the lignin. Can these be calculated and specified?
- 2.) The lignins need to be properly characterized by 2 D NMR methods, as a standard procedure in this field, when looking at depolymerization. The relevant linkage types should be quantified, also the b-O-4 content should be determined as this will have an influence on total theoretical monomer yield obtainable. Then the theoretical yield of dopamine should be given.
- 3.) After depolymerization starting from lignins, the actual monomer yield is typically medium-to good because there are other linkages. The resulting oil is a mixture of monomers, oligomers, condensation products. The manuscript does not mention or quantify any of these products. These are usually detectable by several methods (HPLC, GPC, NMR). Also other monomers than the C2-acetal are expected, but these are not shown or quantified. For better understanding these experiments/analysis should be performed. The yield values reported are unrealistically high, it raises the question of the used analysis method.
- 4.) The authors mention that the C2-aldehyde is unstable. How can the acetal be efficiently deprotected without condensation? Are the yield values reported reliable?

Reviewer #2:

In this manuscript, Dong et al. reported a strategy for production of dopamine from softwood lignin with yield of 74%. This work provides an example of the transformation of renewable biomass waste into high-value chemical products. Their conclusion was characterized by a series of measurements and supported by some deduction. I suggest acceptance of this work after the following points have been taken into account.

The specific comments are listed below.

1. Based on the integration of step 1 and 2 in one-pot, does H-acetal, a by-product of depolymerization of lignin in the first step, affect the conversion of G-acetal into G-ethanol? Perhaps the formation of H-acetal will increase the difficulty of purification of G-ethanol.

2. By products appear in all four-step reactions, and the authors should describe the process of product separation and purification in detail, including solvent recovery, catalyst and additive separation and recovery.
3. Is the 74% dopamine yield obtained from lignin as starting reactant? If not, the four-step reactions should be carried out only with lignin as the starting reactant, and then authors recalculate the yield of dopamine.
4. Sustainable production of dopamine from softwood lignin may be a useful reaction in the future, and the authors should supplement relevant economic and technical analysis.

Reviewer #3:

Remarks to the Author:

The manuscript proposed a new pathway, which includes four individual conversions, for the production of dopamine from lignin. The idea and pathway are important for the valorization of lignin. However, the data and results presented seem insufficient for readers to clearly understand this new pathway. Also, some key results (e.g., the structure and purity of dopamine) must be confirmed by more characterization or tests. Thus, the manuscript is not currently recommended.

(1) the yield of first step and the total yield are strongly suggested to be changed into mass yield (instead of the proportion of theoretic yield), which is clearer for more readers.

(2) I found the pathway and conversion systems proposed in this manuscript are very similar with a recent work of the group of Katalin Barta (DOI: 10.26434/chemrxiv-2022-b6rn4).

(3) Also, the contents in Step 1 (especially, the results in Table 1) had been reported by Barta (e.g., ChemSusChem 2020, 13, 1 – 11). The authors may simplify this part and cite relevant works.

(4) I found the substrates used in the four conversions are fresh chemicals instead of the product of the previous step. So can the authors synthesis dopamine using lignin as the sole substrate?

(5) The final product may not be dopamine. It is probably dopamine hydrobromide since amine group reacts with acid easily and dopamine has a high solubility in aqueous phase so it should not form large amounts of the deposit. The authors are suggested to use GC (after derivatization) or LC to qualitatively and quantitatively analyze the product and compare with dopamine standard. Meanwhile, the purity of the product can be simultaneously determined. The purity (99%) based on NMR signal is not convincing.

(6) What is the method for quantitatively analyzing dopamine?

(7) More details should be provided, for example: 1. the calculation methods for yields (are they based on mass or mol?) 2. the NMR conditions should be provided. The baseline of ¹³C-NMR in Fig. 4 should be shown. 3. what are the concentrations of the acids used in step 4?

(8) "10%Ni/TiO₂ catalyst and excluding the effects of deactivation caused by leaching or aggregation of gold (Fig. 3b)." What does "the gold" mean?

(9) "Then the amination between G-aldehyde and NH₃ occurs spontaneously" Proof should be given for this step under the reaction condition.

Response to Editor and Reviewer –NCOMMS-22-34471A

Comments in blue - Replies in black - Amendments to the manuscript or supporting information in bold. Line and page numbers in the replies refer to the revised manuscript and supporting information with highlighted changes.

We gratefully thank the Editor and Reviewers for their valuable suggestions on this manuscript. We have made efforts to revise the manuscript to further enhance the quality and ensure the highest excellence of our contribution. The revised sections in the manuscript are highlighted in red. We are pleased to answer the questions raised by the Editor and Reviewers point by point as follows:

Reviewer #1

This contribution of Chen and coworkers describes the production of dopamine from lignin. This is an important molecule, and the manuscript is a valuable contribution to the array of strategies that are toward the making of amines from biomass. However i believe it is perhaps more suitable for a specialized journal such as Green Chemistry. Recently, several groups have described the formation of amines from biomass (Tao Zhang, Ning Yan - Angew Chem 2017 on primary amines from biomass, several papers from Sels, Saravanamurugan S 2022 ChemSusChem) and also reviewed by B Sels Green Chem 2017 and these rely mostly on similar catalytic systems and reductive amination or direct amination of alcohols with ammonia. Same time, also the lignin depolymerization field has been investigated extensively in the past years. This manuscript does not describe a novel method for lignin depolymerization or amination.

Response: We thank you for this valuable suggestion. The catalytic synthesis of amines from sustainable biomass is a growing field of interest and the last five years have seen a significant increase in the number of reports and reviews on bio-based amines (*Angew. Chem. Int. Ed.* **56**, 3050-3054 (2017); *Green Chem.* **18**, 487-496 (2016); *Green Chem.* **19**, 5303-5331 (2017); *ChemSusChem* **15**, e202200107 (2022)). However, the substrate scope for the production of bio-based amines is normally limited to (hemi)cellulosic alcohols. Owing to lignin's recalcitrant and complex structure, its conversion into amines remains a formidable challenge. Recently, some literature described the conversion of lignin-derived platform chemicals to amines (*Chem. Sci.* **6**, 4174-4178 (2015); *Nat. Catal.* **1**, 82-92 (2018); *Adv. Synth. Catal.* **362**, 5143-5169 (2020); *Green Chem.* **23**, 6761-6788 (2021)), but the reactants are limited to lignin model compounds rather than natural lignin, as well as complex product distribution, separation difficulty, low value-added, and harsh reaction conditions.

The innate structural features of the aromatic moiety of lignin offer remarkable opportunities for the green manufacturing of essential pharmaceuticals, which represent a large and continuously growing market (*ACS Cent. Sci.* **5**, 1707-1716 (2019); *ChemSusChem* **13**, 5199-5212 (2020); *iScience* **24**, 102211 (2021); *ChemRxiv* (2022) DOI: 10.26434/chemrxiv-2022-b6rn4). Design of synthetic pathways and selective modification of functional groups are essential for this process, more than improving the efficiency of each step (*e.g.*, depolymerization, amination).

In this work, we develop an innovative strategy for the production of dopamine direct from lignin, achieving high-purified dopamine obtained by a simple filtration process. **The novelties of this work have been summarized as follows:**

(1) Compared with other amines, dopamine has higher added value and wider application. Dopamine is one of the most important pharmaceuticals with annual global market revenue of 320 million USD (Cognitive Market Research. Global dopamine market report 2022. <https://www.cognitivemarketresearch.com/dopamine-market-report>), which plays a crucial role in regulating renal function, blood pressure and neurobehavioral disorders (*e.g.*, Alzheimer's disease, Parkinson's disease, schizophrenia, mania, depression, substance abuse, and eating disorders). Moreover, dopamine is also widely used in the field of polydopamine-based materials due to its biocompatibility and self-polymerization capability (**New Supplementary Fig. 1**).

New Supplementary Figure 1. The diverse application pattern of dopamine in pharmaceutical and material synthesis.

(2) To the best of our knowledge, this is the first example of using lignin as the substrate to produce dopamine. According to previous literature, the amines obtained from lignin-derived platform chemicals include aniline, cyclohexylamine, benzylamine, and amphetamine (New Supplementary Fig. 2).

New Supplementary Figure 2. Schematic representation of the conversion of lignin towards bio-based amines based on previous literature.

(3) In comparison with the traditional paths for the production of dopamine, our process has the advantages of high sustainability, use of renewable and cheap materials, high economy, efficient depolymerization, high selectivity, and easy purification (Fig. 1).

(a) Traditional route for dopamine production from vanillin

(b) Traditional route for dopamine production from catechol

(c) Sustainable route for dopamine production from softwood lignin (this work)

Fig. 1 Strategies employed for dopamine production. **a** Traditional route for dopamine production from vanillin. **b** Traditional route for dopamine production from catechol. **c** Lignin to-dopamine route developed in this work.

(4) In this lignin-to-dopamine strategy, lignin was used as the substrate instead of lignin model compounds or lignin-derived platform chemicals. **To further enhance the importance of the study, we have now conducted additional experiments to synthesize dopamine with**

the product from the previous step, obtaining dopamine (3.4 wt.%) with lignin as the sole substrate. It offers a potential protocol for the conversion of native lignin to bio-based dopamine (New Supplementary Fig. 20).

New Supplementary Figure 20. Photos and process yield for the production of dopamine from spruce lignin as the sole substrate.

(5) This work proposes a potential industrial-scale process for the sustainable production of dopamine from softwood lignin. **On the basis of our experimental data, the conceptual design and process modeling, including reaction, product separation, and cyclic utilization, have now been performed (New Fig. 5).** Meanwhile, the techno-economic analysis was also conducted to evaluate economic and environmental potential. The production cost of dopamine in this work was ~2.23 million CNY/t (Chinese Yuan per ton), which was much lower than the dopamine market prices over the last year (4.0~6.0 million CNY/t). The dopamine Prices data are available at the link of <https://www.cognitivemarketresearch.com/dopamine-market-report>.

New Fig. 5 Catalytic conversion of lignin to dopamine. a The yields of the products in the spruce lignin-to-dopamine route. Mass yield = (mass of product) / (mass of lignin) \times 100%; Molar yield = (molar of product) / (mass of lignin); Yield of each step = (molar of product) / (molar of the substrate in each step) \times 100%; Depolymerization efficiency = (yield of G-acetal / theoretical yield of G-acetal) \times 100%; theoretical yield of G-acetal in lignin was obtained by

combining the 2D-HSQC NMR method and derivatization followed by the reductive cleavage method. **b** Conceptual process modeling for dopamine production from lignin. The process model integrates the four catalytic steps: (i) acid-catalyzed lignin depolymerization; (ii) deprotection reaction of G-acetal; (iii) hydrogen-borrowing amination of G-ethanol; and (iv) hydrolysis reaction of G-ethamine into dopamine. **c** Conceptual diagram for dopamine production from lignin. It is comprised of the production and application of dopamine.

The relevant discussion and references have now been updated and cited in the revised manuscript (page 3, lines 2-3, 28-30; page 4, line 1; page 5, lines 4-5; references 24-27).

Some more specific comments and questions are listed below:

Q1. The lignins have been extracted from biomass, however their yield from biomass is not given. The lignin yield from biomass after extraction can be low, so the overall obtained dopamine yield from biomass could be low. The theory monomer yield is also dependent on the β -O-4 content of the lignin. Can these be calculated and specified?

Response: Thanks very much for your suggestion. **We have added the mass yield of lignin from spruce, pine, cedar, and douglas fir in the revised manuscript (page 11, lines 10-11).**

In this work, lignin was extracted from biomass according to a literature procedure (*J. Am. Chem. Soc.* **60**, 1467-1470 (1938); *Nat. Commun.* **8**, 16104 (2017)). The lignin yield of spruce, pine, cedar, and douglas fir was 18.6, 20.1, 17.3, and 16.7 wt.%, respectively, which was nearly the lignin content in softwood (25-30 wt.%) (*Chem. Rev.* **115**, 11559-11624 (2015)).

As suggested in the next comment, **we have carried out both the methods of 2D heteronuclear single-quantum correlation-nuclear magnetic resonance (2D-HSQC NMR) and derivatization followed by reductive cleavage (DFRC) on lignin to calculate the theory monomer yield.**

2D-HSQC NMR method (*Angew. Chem. Int. Ed.* **54**, 258-262 (2015); *J. Am. Chem. Soc.* **137**, 7456-7467 (2015)): The theoretical monomer yield of the acid-catalyzed lignin depolymerization is dependent on the β -O-4 linkages content of the lignin. As shown in the **New Supplementary Fig. 3**, the relative ratios of the main linkages of spruce lignin were determined by 2D-HSQC NMR techniques. Relative quantification of the main linkages provided a β -O-4/ β -5/ β - β ratio of 0.68/0.28/0.04. The H (*p*-hydroxyphenyl)/G (guaiacyl)/S (syringyl) subunit ratio of 0/0.98/0.02, which corresponded well to the H/G/S ratios of product mixtures obtained after lignin depolymerization. In addition, the β -O-4 to monomer ratio was 1/3.0. Based on these data, the theoretical maximum monomer yield from spruce lignin was estimated to be about 11 wt.% assuming that only the β -O-4 linkages were cleaved and

monomeric products were only obtained when two β -O-4 linkages flank a monomer. The theoretical monomer yield of other softwood lignin was also calculated by the same process. The 2D-HSQC NMR characterization of pine lignin, cedar lignin, and douglas fir lignin was conducted and the partial 2D-HSQC NMR spectra were shown in the **New Supplementary Fig. 4**. The β -O-4 to monomer ratio of pine lignin, cedar lignin, and douglas fir lignin was 1/3.3, 1/4.1, and 1/3.4, respectively. Thus, the theoretical maximum monomer yields from pine lignin, cedar lignin, and douglas fir lignin were estimated to be about 9, 6, and 9 wt.%, respectively.

New Supplementary Figure 3 (a) The alkyl region and (b) the aromatic region of the 2D-HSQC NMR spectra of spruce lignin. Some representative structures are shown underneath.

New Supplementary Figure 4 (a-c) The alkyl region and (d-f) the aromatic region of the 2D-HSQC NMR spectra of pine lignin, cedar lignin, and douglas fir lignin. Some representative structures are shown underneath.

DFRC method (*J. Agric. Food Chem.* **45**, 4655-4660 (1997); *ChemSusChem* **13**, 4468-4477 (2020)): In a 10 mL round bottom, 20 mg of lignin and 3 mL of acetyl bromide solution (20/80 AcBr/acetic acid) were added. Then, the mixture was stirred at 130 rpm, 50 °C for 3 hours. The solvent was removed by rotary evaporation at 40 °C and the residue was dissolved in 3 mL stock solution (5/4/1 dioxane/acetic acid/ water). 50 mg zinc dust was added to a solution and stirred for 1 h. The mixture was quantitatively transferred to a separating funnel. Then, 10 mL of DCM, 10 mL of saturated NH₄Cl, and 0.2 mg of tetracosane were added. The pH of the aqueous phase was adjusted to < 3 by adding 3% HCl aqueous solution. The water phase was extracted with 5 mL DCM. Then, the combined DCM fractions were dried over MgSO₄ and the filtrate was evaporated under reduced pressure. The residue was further dissolved in 1.1 mL DCM and 0.4 mL stock solution (1/1 dry pyridine/acetic anhydride) was added under nitrogen. Then, the solution was vortexed and stirred for 1 h. Finally, all volatiles were co-evaporated with ethyl alcohol and the residue was dissolved DCM for GC-FID analysis.

The liquid phase was analyzed by a Shimadzu GC with a HP-5 column (30 m × 0.25 μm)

and run with a temperature profile: 140 °C (held for 1 min), raised at 3 °C/min to 240°C (held for 1 min), raised at 30 °C/min to 300 °C (held for 12 min). Theoretical monomer (G-acetal) yield was calculated using response factors with tetracosane as the internal standard (IS). The calculations were done as follows:

$$m_{G_c \text{ or } G_t} = \text{response factor} \times m_{IS} / \text{Area}_{G_c \text{ or } G_t} \quad (\text{eq. 1})$$

$$\text{mol}_{G_c+G_t} = (m_{G_c} + m_{G_t}) / 264.277 \quad (\text{eq. 2})$$

$$\text{Theoretical yield of G-acetal (wt.\%)} = \text{mol}_{G_c+G_t} \times 210.23 / m_{\text{lignin}} \quad (\text{eq. 3})$$

$$\text{Depolymerization efficiency (\%)} = \text{yield of G-acetal} / \text{theoretical yield of G-acetal} \quad (\text{eq.4})$$

The GC-FID chromatogram of DFRC monomers from spruce lignin and the mass spectrum of G-monomers were shown in the **New Supplementary Fig. 8**. The theoretical monomer yield of spruce lignin is 10.8 wt.%, which is consistent with the results from 2D-HSQC NMR method (~11 wt.%). After the acid-catalytic depolymerization of spruce lignin using ethylene glycol as a stabilization agent, the yield of G-acetal was up to 10.3 wt.% with high depolymerization efficiency (95.4%). The quantitative analysis of the theoretical monomer yield of other softwood lignin was conducted in the same method. The theoretical yield of pine lignin, cedar lignin, and douglas fir lignin was 8.8, 5.4, and 9.5 wt.%, consistent with the 2D-HSQC NMR results (~9, 6, and 9 wt.%). Combined with the yield of G-acetal, the depolymerization efficiency of pine lignin, cedar lignin, and douglas fir lignin was 97.7%, 96.3%, and 83.2%, respectively (**Revised Table 1**).

New Supplementary Figure 8 (a) GC-FID chromatogram of DFRC monomers from spruce lignin and (b) the mass spectrum of G-monomers.

Revised Table 1. Acid-catalytic lignin depolymerization with ethylene glycol stabilization.^a

Entry	Lignin	Yield of G-acetal (wt. %)	Theoretical yield of G-acetal (wt.%) ^b	Depolymerization efficiency (%) ^c
1	Spruce	10.3	10.8	95.4
2	Pine	8.6	8.8	97.7
3	Cedar	5.2	5.4	96.3
4	Douglas fir	7.9	9.5	83.2

^aReaction conditions: lignin (0.2 g), ethylene glycol (0.72 mL), 1,4-dioxane (30 mL), H₂SO₄ (16 μL), temperature (140 °C), 1 h.
^bTheoretical yield of G-acetal was obtained by combining the 2D-HSQC NMR and DFRC methods.
^cDepolymerization efficiency (%) = yield of G-acetal / theoretical yield of G-acetal

The relevant discussion has now been updated in the revised manuscript (page 5, lines 21-26; page 6, lines 30-33) and Supplementary Information (page 3, lines 18-30; page 4, lines 1-30; page 5, lines 1-5).

Q2. The lignins need to be properly characterized by 2 D NMR methods, as a standard procedure in this field, when looking at depolymerization. The relevant linkage types should be quantified, also the b-O-4 content should be determined as this will have an influence on total theoretical monomer yield obtainable. Then the theoretical yield of dopamine should be given.

Response: Thanks for the Reviewer's valuable suggestion. We have collected the 2D-HSQC NMR spectra of lignins from spruce, pine, cedar, and douglas fir to determine the quantities of major linkages and the H/G/S ratio (New Supplementary Fig. 3, 4, and Revised Supplementary Table 1), as discussed in the Response to Question 1. Over spruce lignin, relative quantification of the main linkages provided a β -O-4/ β -5/ β - β ratio of 0.68/0.28/0.04. In addition, the β -O-4 to monomer ratio was 1/3.0. Based on these data, the theoretical maximum monomer yield from spruce lignin was estimated to be about 11 wt.% assuming that only the β -O-4 linkages were cleaved and monomeric products were only obtained when two β -O-4 linkages flank a monomer.

The theoretical yield of dopamine from lignin was calculated by combining the 2D-HSQC NMR and DFRC methods (referred to the Response to Question 1). In the DRFC process, the calculations were done as follows:

$$m_{Gc \text{ or } Gt} = \text{response factor} \times m_{IS} / \text{Area}_{Gc \text{ or } Gt} \quad (\text{eq. 1})$$

$$\text{mol}_{Gc+Gt} = (m_{Gc} + m_{Gt}) / 264.277 \quad (\text{eq. 2})$$

$$\text{The theoretical yield of dopamine (wt.\%)} = \text{mol}_{Gc+Gt} \times 153.18 / m_{\text{lignin}} \quad (\text{eq. 5})$$

Combined with 2D-HSQC NMR and DFRC methods, the theoretical maximum yield of dopamine from spruce lignin, pine lignin, cedar lignin, and douglas fir lignin was 7.9, 6.4, 3.9, and 6.9 wt.%, respectively.

Revised Supplementary Table 1. The content of major linkages and the subunits ratio of different lignin.

Softwood lignin	Ratio of major linkages (%)			Ratio of subunits (%)			Theoretical yield of dopamine (wt.%)
	β -O-4	α -O-4	β - β	S	G	H	
Spruce	68%	28%	4%	0%	98%	2%	7.9
Pine	62%	32%	6%	1%	96%	3%	6.4
Cedar	53%	24%	23%	0%	98%	2%	3.9
Douglas fir	64%	30%	6%	1%	97%	2%	6.9

^aThe data were obtained *via* 2D-HSQC NMR characterization.

“S” refers to syringyl units, “G” refers to guaiacyl units, “H” refers to *p*-hydroxyphenyl units.

Furthermore, we have recorded the 2D-HSQC NMR spectra of the residue after the acid-catalyzed depolymerization of lignin and compared it with that of the raw spruce lignin (New Supplementary Fig. 11). Compared with the NMR signals for the raw spruce lignin (New Supplementary Fig. 3), those for A (β -O-4 linkages), B (β -5 linkages), and C (β - β linkages) decreased significantly in the alkyl regions, and some new signals of AG appeared after the reaction. These results indicate that the structures of the three major linkages (i.e., A, B, and C) were effectively changed due to the cleavage of C-O bonds, and the structures with 1,3-dioxolane end groups were formed. In addition, a large number of G units exist in raw spruce lignin with a hint of H unit. After a reaction for 1h, the G unit vanished completely, and there was a new signal assigned to the AG unit, suggesting that transformation from G to AG using ethylene glycol as the stabilization agent.

New Supplementary Figure 11. (a) The alkyl region and (b) the aromatic region of the 2D-HSQC NMR spectrum of the residue obtained from the acid-catalytic depolymerization of spruce lignin. Some representative structures are shown underneath.

The relevant discussion has now been updated in the revised manuscript (page 5, lines 21-26; page 6, lines 30-33).

Q3. After depolymerization starting from lignins, the actual monomer yield is typically medium-to good because there are other linkages. The resulting oil is a mixture of monomers, oligomers, condensation products. The manuscript does not mention or quantify any of these products. These are usually detectable by several methods (HPLC, GPC, NMR). Also other monomers than the C2-acetal are expected, but these are not shown or quantified. For better understanding, these experiments/analysis should be performed. The yield values reported are unrealistically high, it raises the question of the used analysis method.

Response: Thanks very much for your suggestion. As suggested, **we have carried out a series of characterizations (eg. GPC, 2D-HSQC NMR, GC-MC, and GC-FID) to analyze the resulting lignin oil.** A molecular weight average typical for raw spruce lignin ($M_n = 1947 \text{ g mol}^{-1}$ and $M_w = 4522 \text{ g mol}^{-1}$) was confirmed by Gel Permeation Chromatography (GPC) analysis (**New Supplementary Fig. 10**). After lignin depolymerization, the molecular weight average decreased dramatically ($M_n = 1048 \text{ g mol}^{-1}$ and $M_w = 1789 \text{ g mol}^{-1}$), illustrating that the linkages between subunits in lignin were broken efficiently. The 2D-HSQC NMR spectra of spruce lignin before and after depolymerization were shown in the **New Supplementary Fig.**

3 and 11, confirming that the effective change of main linkages (β -O-4, β -5, and β - β) and the formation of the structures with 1,3-dioxolane end groups (referred to the response to Question 2).

New Supplementary Figure 10. GPC of raw spruce lignin and the resulting lignin oil after depolymerization.

Because of the complexity of the resulting oil, it is difficult to distinguish and qualitative these products using HPLC technology. Here, the mixture products after lignin depolymerization were identified and quantified by GC-MS and GC-FID. As the major product, the yield of G-acetal was up to 10.3 wt.%, as well as the formation of other by-products. These by-products included H-acetal, other monomers and dimers, whose mass yields and structure were summarized in the **New Supplementary Table 3**, and **New Supplementary Fig. 9**. The total mass yield of these by-products was ~1.8 wt.%. Other products with high boiling points could not be detected by GC-FID.

The relevant discussion has now been updated in the revised manuscript (page 7, lines 8-19).

New Supplementary Table 3. The GC-FID chromatogram of lignin oil obtained after spruce lignin depolymerization.

Entry	Ret. (min)	Compound	Yield (wt.%)	Entry	Ret. (min)	Compound	Yield (wt.%)
1	3.035	1,4-dioxane	--	6	21.555		0.5
2	3.740	Ethylene glycol	--	7	22.600		0.2
3	17.061	Pentadecane	--	8	24.105	Dimers	0.3
4	19.061		0.3	9	26.365		0.3
5	20.280		10.3	10	28.315		0.2

Reaction conditions: spruce lignin (0.2 g), ethylene glycol (0.72 mL), 1,4-dioxane (30 mL), H₂SO₄ (16 μL), temperature (140 °C), 1 h.

New Supplementary Figure 9. GC-MS spectra of the products (entries 4, 5, 6, 7, 9, and 10) in Supplementary Table 3.

In order to better understand, the mass yields and molar yields based on lignin, and the yield of each step were all provided in the revised manuscript (New Fig. 5a). After acid-catalyzed depolymerization of spruce lignin, the yield of G-acetal was 10.3 wt.% based on lignin with a 95.4% of depolymerization efficiency (theoretical maximum mass of spruce lignin,

10.8 wt.%). After the deprotection reaction, the yield of G-ethanol was 7.4 wt.%. After the amination reaction, 6.5 wt.% of G-thamine was obtained. Finally, the mass yield of dopamine was 5.7 wt.% based on spruce lignin. By combining 2D-HSQC NMR and DFRC methods, the theoretical maximum mass of dopamine from spruce lignin was determined to be 7.9 wt.%. Though the mass yield of dopamine based on lignin was not high, the high value-added and huge market of dopamine were attractive. The techno-economic analysis was conducted to prove a great potential industrial-scale process for the catalytic conversion of lignin to dopamine. **The relevant discussion has now been updated in the revised manuscript (page 10, lines 6-11).**

New Fig. 5a The yields of the products in the lignin-to-dopamine route. Mass yield = (mass of product) / (mass of lignin) × 100%; Molar yield = (molar of product) / (mass of lignin); Yield of each step = (molar of product) / (molar of the substrate in each step) × 100%; Depolymerization efficiency = (yield of G-acetal / theoretical yield of G-acetal) × 100%; theoretical yield of G-acetal in lignin was obtained by combining the 2D-HSQC NMR method and derivatization followed by the reductive cleavage method.

Q4. The authors mention that the C2-aldehyde is unstable. How can the acetal be efficiently deprotected without condensation? Are the yield values reported reliable?

Response: We appreciate the Reviewer for his/her good question. Based on early literature (*J. Am. Chem. Soc.* **137**, 7456-7467 (2015)), common stabilization agents were diols (*e.g.*, ethylene glycol, 1,3-propanediol, 1,2-butanediol, glycerol) and hydrogen. Particularly, when using hydrogen as the stabilization agent, diols stabilization products (*e.g.*, G-acetal) can be converted into stable alcohols (*e.g.*, G-ethanol). For example, Barta *et al.* reported a catalyst system for the formation of G-ethanol from G-acetal over 5%Ru/Al₂O₃, resulting in a 92% yield at 90 °C with 20 bar hydrogen (*ChemRxiv*, 2022, DOI: 10.26434/chemrxiv-2022-b6rn4, a preprint that was online after the submission of our manuscript). In this work, we presented a process to selectively produce G-ethanol from G-acetal by combining an acid and Ru/C to promote hydrolysis and hydrogenation reactions simultaneously (**New Supplementary Fig. 12**). Without hydrogen and hydrogenation catalyst, the G-aldehyde was easy to condensation in acid condition, generating dimers, trimers, and so on.

New Supplementary Figure 12. Reaction path and mechanism for deprotection of G-acetal into G-ethanol.

In order to prove the reaction path and mechanism, some new control experiments have been conducted and added to the revised manuscript (New Supplementary Table 4). Over only the Ru/C catalyst, the yield of G-ethanol was only 23.3% with 11.5% of the over-hydrogenolysis product (G-ethane). Moreover, the hydrolysis of G-acetal was conducted by adding 1 mL of H₂O at 120 °C for 1h, in which the yield of G-aldehyde was 77.5% with high conversion (80.9%). However, the yield of G-aldehyde decreased significantly from 77.5 to 7.3% with prolonged reaction time (entry 3), attributed to the condensation of G-aldehyde in the presence of acid. Interestingly, both Ru/C and H₂O exist in the reaction system under an H₂/N₂ atmosphere, and the yield of G-ethanol was up to 90.3% with near complete conversion. Particularly, no obvious decrease of G-ethanol with prolonged reaction time (entry 5), suggesting that the G-ethanol was stable in this reaction system. In this strategy, G-acetal was converted into G-aldehyde *via* hydrolysis, followed by hydrogenation to generate stable G-ethanol. **The relevant discussion has now been updated in the revised manuscript (page 8, lines 7-9).**

New Supplementary Table 4. Results for deprotection of G-acetal into G-ethanol.

No.	Reaction condition				Yield (%)			Conv. (%)
	Ru/C (g)	H ₂ /N ₂ (MPa)	H ₂ O (mL)	Time (h)	G-aldehyde	G-ethane	G-ethanol	
1	0.1	1	--	1	0.7	11.5	23.3	43.1
2	--	--	1	1	77.5	0	0	80.9
3	--	--	1	6	7.3	0	0	100
4	0.1	1	1	1	0	0	90.3	99.7
5	0.1	1	1	6	0	0	90.1	100

Reaction conditions: G-acetal (0.02 g), 1,4-dioxane (30 mL), H₂SO₄ (16 uL), 120 °C.

Reviewer #2

In this manuscript, Dong et al. reported a strategy for production of dopamine from softwood lignin with yield of 74%. This work provides an example of the transformation of renewable biomass waste into high-value chemical products. Their conclusion was characterized by a series of measurements and supported by some deduction. I suggest acceptance of this work after the following points have been taken into account.

Response: We warmly thank you for recognizing the novel aspects of our contribution and for supporting its publication in Nature Communications. Your thoughtful suggestions enabled us to strengthen the impact of our work.

The specific comments are listed below.

Q1. Based on the integration of step 1 and 2 in one-pot, does H-acetal, a by-product of depolymerization of lignin in the first step, affect the conversion of G-acetal into G-ethanol? Perhaps the formation of H-acetal will increase the difficulty of purification of G-ethanol.

Response: Thanks very much for your suggestion. **A control experiment using the mixtures of G-acetal and H-acetal as substrates has been conducted.** When the mixtures of G-acetal (90%) and H-acetal (10%) were used as substrates, the yield of G-ethanol and H-ethanol was up to 91.1% and 89.5%, respectively (**Fig. R1**). This result was similar to the result of using G-acetal as the sole substrate (90.3%), suggesting that H-acetal has little influence on the deprotection of G-acetal.

Fig. R1 The deprotection reaction of mixtures of G-acetal and H-acetal over the H₂SO₄ and Ru/C catalyst. Reaction conditions: G-acetal (0.018g), H-acetal (0.002g), Ru/C (0.1g), 1,4-dioxane (30 mL), H₂/N₂ (50 vol%, 1.0 MPa), H₂O (1 mL), H₂SO₄ (16 μL), 120 °C for 1 h.

Except for H-ethanol, other by-products after lignin depolymerization were also detected by GC-FID, and their structure and mass yields were summarized in the **New Supplementary Table 3**. The total mass yield of by-products was about 1.8 wt.%, which would increase the difficulty of purification of G-ethanol. In the laboratory, the method of column chromatography is an effective way for the purification of G-ethanol. In the industrial production process, separation and purification of G-ethanol may be achieved via a rectifying tower.

New Supplementary Table 3. The GC-FID chromatogram of lignin oil obtained after spruce lignin depolymerization.

Entry	Ret. (min)	Compound	Yield (wt.%)	Entry	Ret. (min)	Compound	Yield (wt.%)
1	3.035	1,4-dioxane	--	6	21.555		0.5
2	3.740	Ethylene glycol	--	7	22.600		0.2
3	17.061	Pentadecane	--	8	24.105	Dimers	0.3
4	19.061		0.3	9	26.365		0.3
5	20.280		10.3	10	28.315		0.2

Reaction conditions: spruce lignin (0.2 g), ethylene glycol (0.72 mL), 1,4-dioxane (30 mL), H₂SO₄ (16 μL), temperature (140 °C), 1 h.

Q2. By products appear in all four-step reactions, and the authors should describe the process of product separation and purification in detail, including solvent recovery, catalyst and additive separation and recovery.

Response: Thank you for this critical suggestion. **A tenfold scaling-up experiment with sole spruce lignin as substrate was carried out, in which the process of dopamine synthesis, mass yields of products, product purification, solvent recovery, catalyst, and additive separation were shown in detail in the New Supplementary Fig. 20.**

Firstly, the depolymerization of lignin into G-acetal was conducted in a 250 mL stainless-steel autoclave reactor. Spruce lignin (2.0 g) was mixed with ethylene glycol (7.2 mL), H₂SO₄ (160 μ L), 1,4-dioxane (300 mL) and reacted at 140 °C for 10 h. After the reaction, the reactor was quenched in an ice-water bath and the mixtures were quantitatively analyzed by GC-FID. The yield of G-acetal was 9.5 wt.% based on the lignin. Next, Ru/C (1.0 g) and H₂O (10 mL) were filled into the autoclave, which was sealed and charged to an initial pressure of 1.0 MPa with H₂/N₂ (50 vol%). The autoclave was heated to 120 °C for 10 h. After the deprotection reaction, the mixtures were analyzed by GC-FID and the yield of G-ethanol was up to 6.3 wt.%. The mixtures were centrifuged to separate the liquid phase and Ru/C catalyst, and the latter was recycled for the deprotection reaction. Reduced pressure distillation of the liquid phase was carried out to achieve 1,4-dioxane recovery, which was used for lignin depolymerization again. Further separation of remaining mixtures was conducted by column chromatography, obtaining highly purified G-ethanol. Then, the amination of G-ethanol was tested in a fixed-bed reactor. The 10%Ni/TiO₂ catalyst packed into the tubular reactor was *in-situ* reduced at 400 °C for 4 h. A feed of G-ethanol in *p*-xylene was injected into the fixed-bed reactor with a 0.163 h⁻¹ of weight-hourly space velocity (WHSV). The reactor was heated to 160 °C with an NH₃ flow rate of 100 mL/min. After the hydrogen-borrowing amination reaction, obtained G-ethamine was up to 5.1 wt.%. Reduced pressure distillation of the liquid phase was carried out, followed by the separation of the remaining mixtures. The recycled *p*-xylene solvent was used for the amination reaction and the obtained G-ethamine was further hydrolyzed to produce dopamine in a round-bottom flask. The G-ethamine was mixed with HBr and reacted at 120 °C for 6 h in an oil bath under an N₂ atmosphere. Finally, the dopamine was obtained simply by filtration and drying, yielding 3.4 wt.% based on lignin. The relatively low yield of dopamine might be due to mass loss in the process of transfer, separation, and purification. We are still working on the improvement of the overall efficiency including product purification, solvent recovery, catalyst, and additive separation and recovery.

The relevant discussion has now been updated in the revised manuscript (page 10, lines 10-14) and Supplementary Information (page 5, lines 6-30).

New Supplementary Figure 20. Photos and process yield for the production of dopamine from spruce lignin as the sole substrate.

Q3. Is the 74% dopamine yield obtained from lignin as starting reactant? If not, the four-step reactions should be carried out only with lignin as the starting reactant, and then authors recalculate the yield of dopamine.

Response: Thank you for this critical suggestion. In the original manuscript, the substrates used in the four-step reactions were fresh chemicals and the yield of the product was calculated based on the theoretical maximum yield of lignin. As suggested, **the four-step reactions were carried out only with spruce lignin as the starting substrate (New Supplementary Fig. 20, referred to the Response to Question 2).** With lignin as the sole starting reactant, the yield of dopamine was 3.4 wt.%. Though the mass yield of dopamine based on lignin was not high, the high value-added and huge market of dopamine were attractive. Furthermore, the techno-economic analysis was conducted to prove a great potential industrial-scale process for the catalytic conversion of lignin to dopamine (referred to the Response to the next Question 4).

Q4. Sustainable production of dopamine from softwood lignin may be a useful reaction in the future, and the authors should supplement relevant economic and technical analysis.

Response: Thanks for your valuable suggestion. **On the basis of our experimental data, we have designed a process model to perform a techno-economic analysis to evaluate economic potential.** The process model integrates the four catalytic steps: (i) acid-catalyzed lignin depolymerization; (ii) deprotection reaction of G-acetal; (iii) hydrogen-borrowing

amination of G-ethanol; and (iv) hydrolysis reaction of G-ethamine into dopamine. The conceptual design and process modeling, including reaction, product separation, and solvent recovery, were performed (**New Fig. 5b**). Additionally, the operation of the dopamine plant requires a supporting water supply plant, oil refinery, and electric power plant, in which the production of dopamine from biomass resources can be further applied for industry and pharmaceutical plant (**New Fig. 5c**). The techno-economic analysis of our proposed biorefinery was calculated for annual consumption of 10000 t of lignin. The production cost of dopamine in this work was 2.23 million CNY/t (Chinese Yuan per ton) (**New Supplementary Tables 6-9**), which was much lower than the dopamine market prices in the range between 4.0 and 6.0 million CNY/t over the last year. The dopamine Prices data are available at the link of <https://www.cognitivemarketresearch.com/dopamine-market-report>. The technical economic analysis predicts that this process is an economically competitive production process, and this efficient approach play an important role in lignocellulosic biorefinery in the future. **The relevant discussion has now been updated in the revised manuscript (page 10, lines 15-22).**

New Fig. 5b Conceptual process modeling for dopamine production from lignin. The process model integrates the four catalytic steps: (i) acid-catalyzed lignin depolymerization; (ii) deprotection reaction of G-acetal; (iii) hydrogen-borrowing amination of G-ethanol; and (iv) hydrolysis reaction of G-ethamine into dopamine.

New Fig. 5c Conceptual diagram for dopamine production from lignin. It is comprised of the production and application of dopamine.

New Supplementary Table 6. Assumptions for equipment cost.		
Item description	Equipment cost (CNY)	Equipment cost and other direct costs (CNY)
pump 1	395200	707200
Reactor 1	11719900	14831600
Cooler 1	53700	649100
Makeup H ₂ /N ₂	5565400	7017100
Pump 2	257300	418100
Reactor 2	11913100	15037300
Tower 1	1134900	3154000
Makeup NH ₃	2167600	2830500
Pump 3	271300	446500
Reactor 3	599800	1494063
Tower 2	1134900	3154000
Pump 4	255900	401200
Reactor 4	749400	1904600
Separation	1046900	1722700
total	37265300	53767963

New Supplementary Table 7. Assumptions for electricity cost of equipment.			
Item description	Power (KW)	Electricity (KW·h)	Total cost (CNY)
pump 1	45	360000	180000
Reactor 1	150	1200000	600000
Cooler 1	3	24000	12000
Makeup H ₂ /N ₂	190	1520000	760000
Pump 2	3	24000	12000
Reactor 2	160	1280000	640000
Tower 1	--	--	--
Makeup NH ₃	1.11	8880	4440
Pump 3	7.5	60000	30000
Reactor 3	--	--	--
Tower 2	--	--	--
Pump 4	1.5	12000	6000
Reactor 4	22.7	181600	90800
Separation	15	120000	60000
Total		4790480	2395240

New Supplementary Table 8. Assumptions for raw material cost.			
Items	Unit price (CNY/t)	Quantity(t/year)	Total cost (CNY/year)
Lignin	1000	10000	10000000
Sulfuric acid	514.9851	1464.4	754144.1804
Ethylene glycol	8123.2934	234.9	1908161.62
1,4-dioxane	14900	15703	233974700
process water	15	50000	750000
N ₂	795.6	3500	2784600
H ₂	10000	500	5000000
Ru/C (5 wt.%)	3260400	250	815100000
Paraxylene	6902.2281	1546.3	10672915.31
NH ₃	2438.8256	75.89	185082.4748
Ni/TiO ₂ (10wt%)	121857	1250	152321250
HBr	16000	59.13	946080
Total			1234396934

New Supplementary Table 9. Assumptions for the estimation of the total product cost

Component		Base	Cost (CNY)	
Equipment cost and other direct costs	1.1	Installed cost of all equipment	53767963	
Indirect costs	1.2	50% of 1.1	32260777.8	
Raw materials	2.1	All costs of raw materials and catalysts	1234396934	
Utilities	2.2	Cooling water	3 CNY/t	820800
		Electricity	0.5 CNY/kW·h	2395240
Operating and maintenance cost	2.3.1	Operating labors	9 operators, 100000 CNY/operator/year	900000
	2.3.2	Direct supervisory and clerical labor	20% of (2.3.1)	180000
	2.3.3	Maintenance and repairs	10% of fixed capital	860287.408
	2.3.4	Operating supplies	2% of fixed capital	1720574.816
	2.3.5	Laboratory charge	15% of (2.3.1)	135000
Depreciation	2.4	Straight line depreciation, life period 20y, salvage value 4%	4129379.558	
Plant overhead cost	2.5	60% of 2.3.1+2.3.2+2.3.3	1164172.445	
Administrative cost	2.6	2% of the product cost		
Distribution and selling cost	2.7	2% of the product cost		
Unit price of dopamine (CNY/t)			2235466.351	

Reviewer #3

The manuscript proposed a new pathway, which includes four individual conversions, for the production of dopamine from lignin. The idea and pathway are important for the valorization of lignin. However, the data and results presented seem insufficient for readers to clearly understand this new pathway. Also, some key results (e.g., the structure and purity of dopamine) must be confirmed by more characterization or tests. Thus, the manuscript is not currently recommended.

Response: Thanks very much for your affirming the novel aspects of our contribution. We have added new results and characterizations to illustrate this new pathway in detail. The mass yields of products based on lignin were shown in the revised manuscripts. To explore the feasibility of dopamine synthesis direct from raw lignin, the four-step reactions were also carried out only with lignin as the starting substrate. Particularly, qualitative and quantitative analyses of dopamine were further conducted using more characterizations and tests.

Q1. The yield of first step and the total yield are strongly suggested to be changed into mass yield (instead of the proportion of theoretic yield), which is clearer for more readers.

Response: Thanks for this valuable suggestion. **In order to better understand, the mass yields and molar yields based on lignin, and the yield of each step all were provided in the revised manuscript (New Fig. 5a).** After acid-catalyzed depolymerization of spruce lignin, the yield of G-acetal was 10.3 wt.% based on lignin with a 95.4% of depolymerization efficiency (theoretical maximum mass of spruce lignin, 10.8 wt.%). After the deprotection reaction, the yield of G-ethanol was 7.4 wt.%. After the amination reaction, 6.5 wt.% of G-thamine was obtained. Finally, the mass yield of dopamine was 5.7 wt.% based on spruce lignin. By combining 2D-HSQC NMR and DFRC methods, the theoretical maximum mass of dopamine from spruce lignin was determined to be 7.9 wt.%. Though the mass yield of dopamine based on lignin was not high, the high value-added and huge market of dopamine were attractive. **The relevant discussion has now been updated in the revised manuscript (page 10, lines 6-11).**

New Fig. 5a The yields of the products in the lignin-to-dopamine route. Mass yield = (mass of product) / (mass of lignin) \times 100%; Molar yield = (molar of product) / (mass of lignin); Yield of each step = (molar of product) / (molar of the substrate in each step) \times 100%;

Depolymerization efficiency = (yield of G-acetal / theoretical yield of G-acetal) × 100%; theoretical yield of G-acetal in lignin was obtained by combining the 2D-HSQC NMR method and derivatization followed by the reductive cleavage method.

Q2. I found the pathway and conversion systems proposed in this manuscript are very similar with a recent work of the group of Katalin Barta (DOI: 10.26434/chemrxiv-2022-b6rn4).

Response: Thank you for this important suggestion. The mentioned work from the group of Katalin Barta (DOI: 10.26434/chemrxiv-2022-b6rn4) is a preprint that was online on Oct 18, 2022, which was later than our submission (first submission to Nat. Chem. on Aug. 13, 2022, which was transferred to Nat. Commun. on Aug. 24, 2022) (**Fig. R2**). Thus, this preprint was not cited in our original submission.

Stage	Start Date	End Date	Approximate Duration
Manuscript Transferred	24th August 22		
Decision sent to author	24th August 22		
Manuscript under consideration	16th August 22		
Editor assigned	16th August 22		
Manuscript received	13th August 22		
Manuscript under submission	13th August 22		

Stage	Start Date	End Date	Approximate Duration
	16th December 22		
Manuscript under consideration	16th November 22		
Review Complete	16th November 22		
All Reviewers Assigned	2nd November 22		
Manuscript under consideration	12th September 22		
Contacting Potential Reviewers	12th September 22		
Manuscript under consideration	26th August 22		
Editor Assigned	26th August 22		
Manuscript submitted	24th August 22		
Manuscript Transferred	24th August 22		

Fig. R2 The submission history of the current manuscript to Nat. Chem. on 13th August 2022 and transferred to Nat. Commun. on 24th August 2022.

In this work, we develop an innovative strategy for the production of dopamine direct from lignin. **The novelties of this work have been summarized as follows:** 1) Dopamine has high added value and widespread application, which is one of the most important pharmaceuticals and is widely used in the field of polydopamine-based materials (**New Supplementary Fig. 1**). 2) To the best of our knowledge, this is the first example of using lignin as the substrate to

produce dopamine (**New Supplementary Fig. 2**). 3) In comparison with the traditional paths for the production of dopamine, our process has the advantages of high sustainability, use of renewable and cheap materials, high economy, efficient depolymerization, high selectivity, and easy purification (**Fig. 1**). 4) Lignin was used as the sole substrate instead of lignin model compounds, providing a potential protocol for the conversion of native lignin to bio-based dopamine (**New Supplementary Fig. 20**). 5) On the basis of our experimental data, the conceptual design and process modeling, including reaction, product separation, and cyclic utilization, have now been performed (**New Figs. 5b and 5c**). Meanwhile, the techno-economic analysis was conducted to evaluate economic and environmental potential. This work proposes a potential industrial-scale process for the sustainable production of dopamine from softwood lignin.

Compared to this preprint, our work presents unique novelties: (1) The lignin depolymerization (step 1) and deprotection (step 2) were carried out in one-pot in our work, just adding a little water and catalyst in the second step, therefore reducing the cost of separation and purification. In contrast, it can not be realized in one-pot in Barta's work. (2) Hydrogen-borrowing amination reaction was tested in a fixed-bed reactor in our work. The optimal catalyst was 10% Ni/TiO₂ with the complete conversion with a yield of G-ethamine up to 88.2%. In Barta's work, Raney Ni was selected as the catalyst and the optimal yield of G-ethamine was 68%. (3) The target product was different in these two studies. We aim to produce dopamine, whereas Barta's work aimed to get tetrahydro papaveroline. Therefore, our work exhibits significant advances. **We have better highlighted our novelties (page 3, lines 2-3, 28-30; page 4, line 1; page 5, lines 4-5) and also cited the mentioned preprint (reference 33) in the revised manuscript.**

New Supplementary Figure 1. The diverse application pattern of dopamine in pharmaceutical and material synthesis.

New Supplementary Figure 2. Schematic representation of the conversion of lignin towards bio-based amines based on previous literature.

New Supplementary Figure 20. Photos and process yield for the production of dopamine from spruce lignin as the sole substrate.

New Fig. 5 Catalytic conversion of lignin to dopamine. a The yields of the products in the spruce lignin-to-dopamine route. Mass yield = (mass of product) / (mass of lignin) \times 100%; Molar yield = (molar of product) / (mass of lignin); Yield of each step = (molar of product) / (molar of the substrate in each step) \times 100%; Depolymerization efficiency = (yield of G-acetal / theoretical yield of G-acetal) \times 100%; theoretical yield of G-acetal in lignin was obtained by

combining the 2D-HSQC NMR method and derivatization followed by the reductive cleavage method. **b** Conceptual process modeling for dopamine production from lignin. The process model integrates the four catalytic steps: (i) acid-catalyzed lignin depolymerization; (ii) deprotection reaction of G-acetal; (iii) hydrogen-borrowing amination of G-ethanol; and (iv) hydrolysis reaction of G-ethamine into dopamine. **c** Conceptual diagram for dopamine production from lignin. It is comprised of the production and application of dopamine.

Q3. Also, the contents in Step 1 (especially, the results in Table 1) had been reported by Barta (e.g., *ChemSusChem* 2020, 13, 1 – 11). The authors may simplify this part and cite relevant works.

Response: Thank you for your good suggestion. **The contents in step 1 have been simplified in the revised manuscript (Revised Table 1), in which solvent and aid screening for lignin depolymerization have been moved to Supplement Information (Supplementary Table 2) and more relevant works have been cited (*J. Am. Chem. Soc.* 137, 7456-7467 (2015); *ChemSusChem* 9, 2974-2981 (2016); *ChemSusChem* 13, 4468-4477 (2020)).** In order to better understand, the mass yield of G-acetal based on lignin was also added (**Revised Table 1**). The mass yield of G-acetal from spruce lignin, pine lignin, cedar lignin, and douglas fir lignin was 10.3, 8.6, 5.2, and 7.9 wt.%, respectively.

Revised Table 1 Acid-catalytic lignin depolymerization with ethylene glycol stabilization. ^a				
Entry	Lignin	Yield of G-acetal (wt. %)	Theoretical yield of G-acetal (wt.%) ^b	Depolymerization efficiency (%) ^c
1	Spruce	10.3	10.8	95.4
2	Pine	8.6	8.8	97.7
3	Cedar	5.2	5.4	96.3
4	Douglas fir	7.9	9.5	83.2
^a Reaction conditions: lignin (0.2 g), ethylene glycol (0.72 mL), 1,4-dioxane (30 mL), H ₂ SO ₄ (16 μL), temperature (140 °C), 1 h. ^b Theoretical yield of G-acetal was obtained by combining the 2D-HSQC NMR and DFRC methods. ^c Depolymerization efficiency (%) = yield of G-acetal / theoretical yield of G-acetal				

Supplementary Table 2 Catalyst and solvent screening for G-C2-acetal production in lignin acidolysis with ethylene glycol stabilization.^a

Entry	Lignin	Catalyst	Solvent	Yield of G-acetal (wt. %)	Depolymerization efficiency (%) ^b
1	Spruce	H ₂ SO ₄	1,4-dioxane	10.3	95.4
2	Spruce	H ₂ SO ₄	toluene	1.4	13.0
3	Spruce	H ₂ SO ₄	methanol	3.6	33.3
4	Spruce	H ₂ SO ₄	dimethyl carbonate	6.4	59.3
5	Spruce	TfOH	1,4-dioxane	10.1	93.5
6	Spruce	HCl	1,4-dioxane	0	0
7	Spruce	HNO ₃	1,4-dioxane	0	0
8	Spruce	HF	1,4-dioxane	0	0

^aReaction conditions: spruce lignin (0.2 g), ethylene glycol (0.72 mL), solvent (30 mL), acid (16 μ L), temperature (140 °C), 1 h.

^bDepolymerization efficiency (%) = yield of G-acetal / theoretical yield of G-acetal. Theoretical yield of G-acetal in spruce lignin was 10.8 wt.%, obtained by combining the 2D-HSQC NMR method and derivatization followed by the reductive cleavage method.

The theoretical G-acetal yield is calculated by combining the 2D heteronuclear single-quantum correlation-nuclear magnetic resonance (2D-HSQC NMR) method and derivatization followed by the reductive cleavage (DFRC) method.

2D-HSQC NMR method (*Angew. Chem. Int. Ed.* **54**, 258-262 (2015); *J. Am. Chem. Soc.* **137**, 7456-7467 (2015)): Theoretical monomer yield of the acid-catalyzed lignin depolymerization is dependent on the β -O-4 linkages content of the lignin. As shown in the **New Supplementary Fig. 3**, the relative ratios of the main linkages of spruce lignin were determined by 2D-HSQC NMR techniques. Relative quantification of the main linkages provided a β -O-4/ β -5/ β - β ratio of 0.68/0.28/0.04. The β -O-4 to monomer ratio was 1/3.0. Based on these data, the theoretical maximum monomer yield from spruce lignin was estimated to be about 11 wt.% assuming that only the β -O-4 linkages were cleaved and monomeric products were only obtained when two β -O-4 linkages flank a monomer. Using the same method, the theoretical maximum monomer yields from pine lignin, cedar lignin, and douglas fir lignin were estimated to be about 9, 6, and 9 wt.%, respectively.

New Supplementary Figure 3 (a) The alkyl region and (b) the aromatic region of the 2D-HSQC NMR spectra of spruce lignin. Some representative structures are shown underneath.

DFRC method (*J. Agric. Food Chem.* **45**, 4655-4660 (1997); *ChemSusChem* **13**, 4468-4477 (2020)): Theoretical monomer (G-acetal) yield was calculated using response factors with tetracosane as the internal standard (IS). The calculations were done as follows:

$$m_{Gc \text{ or } Gt} = \text{response factor} \times m_{IS} / \text{Area}_{Gc \text{ or } Gt} \quad (\text{eq. 1})$$

$$\text{mol}_{Gc+Gt} = (m_{Gc} + m_{Gt}) / 264.277 \quad (\text{eq. 2})$$

$$\text{The theoretical yield of G-acetal (wt.\%)} = \text{mol}_{Gc+Gt} \times 210.23 / m_{\text{lignin}} \quad (\text{eq. 3})$$

$$\text{Depolymerization efficiency (\%)} = \text{yield of G-acetal} / \text{theoretical yield of G-acetal (eq.4)}$$

The GC-FID chromatogram of DFRC monomers from spruce lignin and the mass spectrum of G-monomers were shown in the **New Supplementary Fig. 8**. The theoretical monomer yield of spruce lignin is 10.8 wt.%, which is consistent with the results from 2D-HSQC NMR method (~11 wt.%). After the acid-catalytic depolymerization of spruce lignin using ethylene glycol as a stabilization agent, the yield of G-acetal was up to 10.3 wt.% with high depolymerization efficiency (95.4%). The quantitative analysis of the theoretical monomer yield of other softwood lignin was conducted in the same method. The theoretical yield of pine lignin, cedar lignin, and douglas fir lignin was 8.8, 5.4, and 9.5 wt.%, consistent with the 2D-HSQC NMR results (~9, 6, and 9 wt.%). Combined with the yield of G-acetal, the depolymerization efficiency of pine lignin, cedar lignin, and douglas fir lignin was 97.7%, 96.3%, and 83.2%, respectively (**Revised Table 1**).

New Supplementary Figure 8 (a) GC-FID chromatogram of DFRC monomers from spruce lignin and (b) the mass spectrum of G-monomers.

Q4. I found the substrates used in the four conversions are fresh chemicals instead of the product of the previous step. So can the authors synthesis dopamine using lignin as the sole substrate?

Response: Thank you for this critical suggestion. **A tenfold scaling-up experiment with sole spruce lignin as substrate was carried out, in which the process of dopamine synthesis, mass yields of products, product purification, solvent recovery, catalyst, and additive separation in detail were shown in the New Supplementary Fig. 20.**

Firstly, the depolymerization of lignin into G-acetal was conducted in a 250 mL stainless-steel autoclave reactor. Spruce lignin (2.0 g) was mixed with ethylene glycol (7.2 mL), H_2SO_4 (160 μ L), 1,4-dioxane (300 mL) and reacted at 140 $^{\circ}C$ for 10 h. After the reaction, the reactor was quenched in an ice-water bath and the mixtures were quantitatively analyzed by GC-FID. The yield of G-acetal was 9.5 wt.% based on the lignin. Next, Ru/C (1.0 g) and H_2O (10 mL) were filled into the autoclave, which was sealed and charged to an initial pressure of 1.0 MPa with H_2/N_2 (50 vol%). The autoclave was heated to 120 $^{\circ}C$ for 10 h. After the deprotection reaction, the mixtures were analyzed by GC-FID and the yield of G-ethanol was up to 6.3 wt.%. The mixtures were centrifuged to separate the liquid phase and Ru/C catalyst, and the latter was recycled for the deprotection reaction. Reduced pressure distillation of the liquid phase was carried out to achieve 1,4-dioxane recovery, which was used for lignin depolymerization again. Further separation of remaining mixtures was conducted by column chromatography, obtaining highly purified G-ethanol. Then, the amination of G-ethanol was tested in a fixed-bed reactor. The 10%Ni/TiO₂ catalyst packed into the tubular reactor was *in-situ* reduced at 400 $^{\circ}C$ for 4 h. A feed of G-ethanol in *p*-xylene was injected into the fixed-bed reactor with a 0.163 h⁻¹ of weight-hourly space velocity (WHSV). The reactor was heated to 160 $^{\circ}C$ with an NH_3 flow rate

of 100 mL/min. After the hydrogen-borrowing amination reaction, obtained G-ethamine was up to 5.1 wt.%. Reduced pressure distillation of the liquid phase was carried out, followed by the separation of the remaining mixtures. The recycled *p*-xylene solvent was used for the amination reaction and the obtained G-ethamine was further hydrolyzed to produce dopamine in a round-bottom flask. The G-ethamine was mixed with HBr and reacted at 120 °C for 6 h in an oil bath under an N₂ atmosphere. Finally, the dopamine was obtained simply by filtration and drying, yielding 3.4 wt.% based on lignin. The relatively low yield of dopamine might be due to mass loss in the process of transfer, separation, and purification. We are still working on the improvement of the overall efficiency including product purification, solvent recovery, catalyst, and additive separation and recovery.

The relevant discussion has now been updated in the revised manuscript (page 10, lines 11-15) and Supplementary Information (page 5, lines 6-30).

New Supplementary Figure 20. Photos and process yield for the production of dopamine from spruce lignin as the sole substrate.

On the basis of our experimental data, we have designed a process model to perform a techno-economic analysis to evaluate economic potential. The process model integrates the four catalytic steps: (i) acid-catalyzed lignin depolymerization; (ii) deprotection reaction of G-acetal; (iii) hydrogen-borrowing amination of G-ethanol; and (iv) hydrolysis reaction of G-ethamine into dopamine. The conceptual design and process modeling, including reaction, product separation, and solvent recovery, were performed (New Fig. 5b). Additionally, the operation of the dopamine plant requires a supporting water supply plant, oil refinery, and

electric power plant, in which the production of dopamine from biomass resources can be further applied for industry and pharmaceutical plant (**New Fig. 5c**). The techno-economic analysis of our proposed biorefinery was calculated for annual consumption of 10000 t of lignin. The production cost of dopamine in this work was 2.23 million CNY/t, which was much lower than the dopamine market prices in the range between 4.0 and 6.0 million CNY/t over the last year. (The dopamine Prices data sets. <https://www.cognitivemarketresearch.com/dopamine-market-report>). The technical economic analysis predicts that this process is an economically competitive production process, and this efficient approach play an important role in lignocellulosic biorefinery in the future.

New Fig. 5b Conceptual process modeling for dopamine production from lignin. The process model integrates the four catalytic steps: (i) acid-catalyzed lignin depolymerization; (ii) deprotection reaction of G-acetal; (iii) hydrogen-borrowing amination of G-ethanol; and (iv) hydrolysis reaction of G-ethamine into dopamine.

New Fig. 5c Conceptual diagram for dopamine production from lignin. It is comprised of the production and application of dopamine.

Q5. The final product may not be dopamine. It is probably dopamine hydrobromide since amine group reacts with acid easily and dopamine has a high solubility in aqueous phase so it should not form large amounts of the deposit. The authors are suggested to use GC (after derivatization) or LC to qualitatively and quantitatively analyze the product and compare with dopamine standard. Meanwhile, the purity of the product can be simultaneously determined. The purity (99%) based on NMR signal is not convincing.

Response: Thank you for your good suggestion. Indeed, the obtained white powder was dopamine hydrobromide after the hydrolysis of G-ethamine over the HBr catalyst. In HBr (40.0 wt.%) solvent, dopamine was easy to react with HBr, generating dopamine hydrobromide as a solid settling down. Dopamine hydrobromide has a high solubility in water but does not dissolve in high-concentration HBr solvent. However, dopamine hydrochloride and dopamine hydriodate have high solubility in HCl and HI solvents. Thus, the optimal catalyst was HBr, which is beneficial to the separation of products.

In the original manuscript, the yield of dopamine was calculated based on the solid mass. The hydrolysis of G-ethamine over HBr and the quantitative method based on solid mass were also reported in Katalin Barta's work (DOI: [10.26434/chemrxiv-2022-b6rn4](https://doi.org/10.26434/chemrxiv-2022-b6rn4)). In order to give a more accurate quantification, **we have added a new quantitative method using high-performance liquid chromatography (HPLC) technology by an external standard method.** After the hydrolysis reaction, white powder was obtained by filtration and drying, and was

further dissolved in water. The aqueous phase was qualitatively analyzed by HPLC (Agilent 1200 Series) equipped with an Agilent C18 column (Zorbax SB-C18; 4.6 mm × 150 mm, 3.5 μm) and a diode-array detector. 40 μL of the reaction mixture was taken out in a centrifuge tube, 960 μL of water was added to the mixture, and the diluted liquid was filtered through a 0.22 μm filter. 10 μL of the sample was injected under the following conditions: column temperature = 30 °C, mobile phase = acetonitrile (20% v/v) with water (80% v/v), and flow rate = 0.5 mL min⁻¹, wavelengths were chosen 280 nm to observe products. Based on the HPLC analysis (**New Supplementary Fig. 18**), the yield of dopamine was 94.5% over HBr, slightly lower than the mass analyst method (97.0%). In addition, nearly no other product was detected in HPLC, suggesting high purity of dopamine. **The relevant discussion has now been updated in the revised manuscript (page 12, lines 26-30; page 13, lines 1-4).**

New Supplementary Figure 18. Representative HPLC chromatographs of dopamine standard and reaction mixture after hydrolysis of G-ethamine recorded at 280 nm. Reaction condition: G-ethamine (1.0 g), HBr (6.0 g), N₂ pressure (0.1 MPa), temperature (120 °C), 6h.

Q6. What is the method for quantitatively analyzing dopamine?

Response: Thank you very much. In the original manuscript, the quantitative analysis was based on the mass of the white solid. As suggested, **we have added a new quantitative method using high-performance liquid chromatography (HPLC) technology by an external standard method**, which can be referred to the Response to Question 5.

Q7. More details should be provided, for example: 1. the calculation methods for yields (are they based on mass or mol?) 2. the NMR conditions should be provided. The baseline of ¹³C-NMR in Fig. 4 should be shown. 3. what are the concentrations of the acids used in step 4?

Response: Thanks for your valuable suggestion. **We have now provided more details as**

follows:

(1) The mass yields and the molar yields based on lignin, and the yield of each step were all provided in New Fig. 5a, and the calculation methods were shown in the caption.

New Fig. 5a The yields of the products in the lignin-to-dopamine route. Mass yield = (mass of product) / (mass of lignin) \times 100%; Molar yield = (molar of product) / (mass of lignin); Yield of each step = (molar of product) / (molar of the substrate in each step) \times 100%; Depolymerization efficiency = (yield of G-acetal / theoretical yield of G-acetal) \times 100%; theoretical yield of G-acetal in lignin was obtained by combining the 2D-HSQC NMR method and derivatization followed by the reductive cleavage method.

(2) NMR spectra were recorded on Bruker Avance-400 instrument, calibrated to D(H)₂O as the internal reference (4.69 ppm for ¹H NMR spectra). **The baseline of ¹³C-NMR has been shown in the revised manuscript (revised Fig. 4c).**

Revised Fig. 4c ¹H and ¹³C NMR spectra for the obtained dopamine from hydrolysis of G-ethanmine.

(3) The concentrations of the acids used in hydrolysis are shown as follows: HCl (36.0-38.0 wt.%, Sinopharm Chemical), H₂SO₄ (95.0-98.0 wt.%, Sinopharm Chemical), and HBr (40.0 wt.%, Aladdin). All acids were used as received without further attenuation. **We have added these contents in the revised manuscript (page 23, lines 5-6).**

Q8. “10%Ni/TiO₂ catalyst and excluding the effects of deactivation caused by leaching or aggregation of gold (Fig. 3b).” What does “the gold” mean?

Response: We apologize for the typo, the sentence should be: “10%Ni/TiO₂ catalyst and excluding the effects of deactivation caused by leaching or aggregation of nickel”. **We have corrected this in the revised manuscript (page 9, lines 1-2).**

Q9. “Then the amination between G-aldehyde and NH₃ occurs spontaneously” Proof should be given for this step under the reaction condition.

Response: Thanks for your valuable suggestion. According to early literature (*J. Catal.* **378**, 392-401 (2019)), the noncatalytic reaction between ketones (or aldehydes) and NH₃ occurs spontaneously before reaching the chemical balance. The reaction network in hydrogen-borrowing amination of G-aldehyde is shown in the **New Supplementary Figure 17**.

Furthermore, we have carried out the amination reaction between G-aldehyde and NH₃ without a catalyst at 50 °C for 4h, a milder reaction condition than hydrogen-borrowing amination of G-ethanol. As shown in **Fig. R3**, the conversion of G-aldehyde was up to 80.1% and the yield of G-ethyliminium was 12.6%, as well as the formation of lots of dimers. It could be explained that G-ethyliminium was unstable and easy to polymerize into oligomers. This result demonstrates that the amination between G-aldehyde and NH₃ occurs spontaneously.

New Supplementary Figure 17. The reaction network in hydrogen-borrowing amination of G-aldehyde.

Fig. R3 The amination reaction between G-aldehyde and NH₃ without catalysts.

Reviewers' Comments:

Reviewer #2:

Remarks to the Author:

The authors have addressed all the comments very carefully. I believe their response is reasonable and acceptable.

Reviewer #3:

Remarks to the Author:

The revised manuscript is improved. However, some of my major concerns (including the structure and purity of dopamine) were not well-addressed. Thus, the current manuscript needs a major revision.

(1) As the authors clarified, the final product is "dopamine hydrobromide" (it formed in HBr solution). Thus, it is not suitable to name the target product simply as "dopamine" throughout the manuscript.

(2) Although dopamine, dopamine hydrochloride, dopamine hydrobromide, dopamine hydroiodide and other dopamine species can be named as "dopamine", they differ in properties and applications. As I know, the majority of dopamine species applied industrially and commercially is dopamine hydrochloride. For example, dopamine hydrochloride is a widely-used medicine officially registered in China, while dopamine and dopamine hydrobromide are not. Thus, the application of the product of this work, dopamine hydrobromide, may be strongly limited.

(3) The purity is an important parameter of the product and the pathway. The authors should not simply mention that the product is of "high purity", instead, an accurate data of purity should be given. Also, can the authors quantitatively identify the impurities in the product (especially the product from the tandem conversions of lignin)?

(4) Some details of the conversion procedure are still missing. For example, characterization techniques, such as TEM, XRD, ICP...; details in tenfold scaling-up experiments, such as product separation conditions.

(5) The production cost is calculated to be 2.23 million CHY/t, which is still very high to me. The author can give a brief expectation on reducing the cost.

(6) The words "Step 3" is missing in Fig. 5b.

(7) Citations should be given for Fig. S1 and S2.

Response to Editor and Reviewer –NCOMMS-22-34471A

Comments in blue - Replies in black - Amendments to the manuscript or supporting information in **bold**. Line and page numbers in the replies refer to the revised manuscript and supporting information with highlighted changes.

We gratefully thank the Editor and Reviewers for their valuable suggestions on this manuscript. We have made efforts to revise the manuscript to further enhance the quality and ensure the highest excellence of our contribution. The revised sections in the manuscript are highlighted in red. We are pleased to answer the questions raised by the Editor and Reviewers point by point as follows:

Reviewer #1

In private comments to the editorial office, other referee(s) expressed that this reviewer's concerns were addressed appropriately.

Response: We are grateful to the Reviewer and Editorial Office for their positive feedback on the contents of the manuscript.

Reviewer #2

The authors have addressed all the comments very carefully. I believe their response is reasonable and acceptable.

Response: We warmly thank the Reviewer for recognizing the novel aspects of our contribution and for supporting its publication in Nature Communications.

Reviewer #3

The revised manuscript is improved. However, some of my major concerns (including the structure and purity of dopamine) were not well-addressed. Thus, the current manuscript needs a major revision.

Response: We are grateful for the Reviewer's constructive comments and valuable suggestions. **We have conducted additional experiments and characterizations to relieve the concerns about the structure and purity of dopamine.** Importantly, the conversion of dopamine hydrobromide into dopamine hydrochloride was carried out in ethanol with a yield of 92.6 %. The structure and purity of the obtained dopamine hydrochloride were further verified by a series of methods (*vide infra*). Your thoughtful suggestions enabled us to further strengthen the

impact of our work.

Q1. As the authors clarified, the final product is “dopamine hydrobromide” (it formed in HBr solution). Thus, it is not suitable to name the target product simply as “dopamine” throughout the manuscript.

Response: Thanks for this valuable suggestion. **We agree with the Reviewer and have now described the product more accurately with dopamine hydrobromide or dopamine hydrochloride throughout the manuscript.**

Q2. Although dopamine, dopamine hydrochloride, dopamine hydrobromide, dopamine hydroiodide and other dopamine species can be named as “dopamine”, they differ in properties and applications. As I know, the majority of dopamine species applied industrially and commercially is dopamine hydrochloride. For example, dopamine hydrochloride is a widely-used medicine officially registered in China, while dopamine and dopamine hydrobromide are not. Thus, the application of the product of this work, dopamine hydrobromide, may be strongly limited.

Response: Thank you for this critical suggestion. Compared with dopamine hydrobromide, dopamine hydrochloride indeed has a wider application and higher value in the field of medicine. **Therefore, we have carried out additional experiments to convert the obtained dopamine hydrobromide into dopamine hydrochloride, in which the processes of dopamine hydrochloride synthesis, product separation, and qualitative and quantitative analysis of the product were shown in detail in the revised manuscript (page 9, lines 15-22; page 12, lines 23-30; page 13, lines 1-8) and Supplementary Information (page 6, lines 7-34; page 7, lines 1-20).**

The conversion of dopamine hydrobromide into dopamine hydrochloride was carried out in a 10 mL vial (**Revised Fig. 4c**). Firstly, dopamine hydrobromide (1.0 g) was mixed with ethanol (8 mL) under an N₂ atmosphere at 20 °C, which dissolved most of the dopamine hydrobromide. Then, the mixture was heated to 80 °C in an oil bath and the dopamine hydrobromide was found to be completely soluble in ethanol. Next, concentrated HCl (36.0-38.0 wt.%, 1 mL) was slowly dripped into the solution. Afterward, the system was cooled to 0 °C in an ice-water bath, and a large amount of the targeted dopamine hydrochloride could be obtained in the form of white powder precipitates at the bottom of the vial. Finally, the white powder could be easily separated by a simple filtration and drying process and was further analyzed by a series of methods.

The relevant discussion has now been updated in the revised manuscript (page 9, lines 15-21; page 12, lines 23-30).

Revised Fig. 4 Catalytic hydrolysis of G-ethamine to dopamine hydrochloride and product separation. **a** Digital photo of the reaction mixtures after the hydrolysis of G-ethamine into dopamine hydrobromide in an aqueous solution over different acids. **b** Catalytic results for the hydrolysis of a G-ethamine in an aqueous solution over different acids. Reaction condition: G-ethamine (1.0 g), acid (6.0 g), N₂ pressure (0.1 MPa), temperature (120 °C). The concentrations of HBr, HI, and HCl were 40.0 wt.%, 55.0-58.0 wt.%, and 36.0-38.0 wt.%, respectively. **c** Digital photo of the reaction mixtures after the conversion of dopamine hydrobromide into dopamine hydrochloride in ethanol. **d** ¹H and ¹³C NMR spectra for the obtained dopamine hydrochloride from Fig. 4c. NMR spectra were recorded on Bruker Avance-400 instrument, using D₂O as solvent.

According to Chinese Pharmacopoeia, **the obtained white powder in ethanol was qualitatively and quantitatively analyzed via a series of methods**

(<https://db.ouryao.com/yd2020/view.php?id=fc6dfd4025>), including the color test method, ultraviolet-visible spectroscopy (UV-Vis), Fourier transform infrared spectroscopy (FT-IR), mass spectrometry (MS), nuclear magnetic resonance (NMR), high-performance liquid chromatography (HPLC), ion chromatography (IC), and titrimetric analysis method.

a) Color test method: The white powder (10 mg) was first dissolved in deionized water (1 mL) forming a colorless solution, which changed to dark green after dripping FeCl₃ solution. The color of the solution was further changed to purplish red after adding NH₃·H₂O (**New Supplementary Fig. 20a**).

b) UV-Vis method: The white powder (3 mg) was first dissolved in H₂SO₄ solution (0.5 wt.%, 100 mL) and was further measured by UV-Vis spectroscopy. As shown in **New Supplementary Fig. 20b**, a single peak at 280 nm assigned to the benzene ring group was observed, agreeing well with the commercial dopamine hydrochloride standard.

c) FT-IR method: The white powder (1 mg) was mixed with dried KBr (0.2 g) and finely ground in an agate mortar, which was further pressed into a transparent slice for FT-IR analysis. As shown in **New Supplementary Fig. 20c**, the white powder presented characteristic bands at 1286, 1499, 1612, 2953, 3041, 3216-3343 cm⁻¹ assigning to the stretching of C-O, C-C (aromatic), C-N, C-H (alkyl), C-H (aromatic), and -OH groups, respectively. The FT-IR spectrum of white powder was nearly the same as that of the commercial dopamine hydrochloride, suggesting that the white powder should be dopamine hydrochloride.

d) MS method: As shown in **New Supplementary Fig. 20d**, the mass spectrum of the white powder was consistent with that of the commercial dopamine hydrochloride, further proving the structure of white powder. MS (ESI): 154 [M+H]⁺.

e) NMR method: ¹H and ¹³C NMR spectra for the obtained white powder were shown in **Revised Fig. 4d**, which were consistent with the results of the previous studies (*Bull. Chem. Soc. Jpn.* **63**, 1252-1254 (1990)). ¹H NMR (400 MHz, D₂O): δ 6.86 (d, *J* = 8.0 Hz, 1H), 6.81 (s, 1H), 6.72 (d, *J* = 8.0 Hz, 1H), 3.19 (t, *J* = 7.0 Hz, 2H), 2.84 (t, *J* = 7.0 Hz, 2H). ¹³C NMR (101 MHz, D₂O): δ 144.1, 142.9, 129.2, 121.1, 116.5, 116.4, 40.6, 31.9.

f) HPLC method: In most of the previous literature (*J. Pharm. Biomed. Anal.* **21**, 519-525 (1999); *Synthesis*, **22**, 3838-3842 (2009); *J. Am. Chem. Soc.* **133**, 6948-6951 (2011)), the yield of dopamine hydrochloride was determined by HPLC analysis. Therefore, the yield and purity of the white powder were also determined by HPLC (Agilent 1200 Series), equipped with an Agilent C18 column (Zorbax SB-C18; 4.6 mm × 150 mm, 3.5 μm) and a diode-array detector. The mobile phase was sodium dodecyl sulfate (0.005 mol/L)/acetonitrile/glacial acetic acid/ethylenediaminetetraacetic acid disodium salt solution (0.1 mol/L) = 700/300/10/2. In brief,

the obtained white powder was dissolved in the mobile phase (0.3 g/L). 10 μ L of the sample solution was injected under the following conditions: column temperature = 30 °C, flow rate = 1.0 mL min⁻¹, and wavelength = 280 nm. As shown in **New Supplementary Fig. 20e**, a peak at 10.9 min was observed in the corresponding HPLC chromatograph, consisting with the commercial dopamine hydrochloride. The yield of dopamine hydrochloride was calculated to be 92.6% using 4-ethylpyrocatechol as the internal standard. Further combined with ion chromatography (IC) results, the purity of obtained dopamine hydrochloride was determined to be 98.0 % and the major impurities were 4-(2-aminoethyl)-2-methoxyphenol hydrochloride (0.9%) and dopamine hydrobromide (0.8%).

g) Titrimetric analysis method: The obtained white powder (150 mg) was first mixed with glacial acetic acid (25 mL). After boiling at 120 °C for 5 mins, the solution was cooled down to 40 °C. Next, mercury acetate test solution (5 mL) and a drop of leucocrystal violet were added to the solution. Finally, the solution was titrated with perchloric acid titrant (0.1 mol/L) until the color changed to green and corrected the result with a blank test. 1 mL of perchloric acid titrant (0.1 mol/L) is equivalent to 18.96 mg of dopamine hydrochloride. Based on titrimetric analysis, the purity of obtained dopamine hydrochloride was 98.2%, similar to results from the HPLC method (98.0%).

The relevant discussion has now been updated in the revised manuscript (page 9, lines 21-22; page 13, lines 1-8) and Supplementary Information (page 6, lines 7-34; page 7, lines 1-20).

New Supplementary Figure 20. The qualitative and quantitative analysis of the obtained dopamine hydrochloride from dopamine hydrobromide in ethanol. **a** Pictures for the color test of obtained dopamine hydrochloride aqueous solution. **b** The UV-Vis spectra of the obtained and commercial dopamine hydrochloride. **c** The FT-IR spectra of the obtained and commercial dopamine hydrochloride. **d** The MS of obtained dopamine hydrochloride. **e** The representative HPLC chromatographs of the obtained and commercial dopamine hydrochloride recorded at 280 nm.

In the conversion of spruce lignin into dopamine hydrochloride, the mass yields and molar yields based on lignin, and the yield of each step were summarized in the revised Fig. 5a. After acid-catalyzed depolymerization of spruce lignin, the mass yield of G-acetal was

10.3 wt.% with a depolymerization efficiency of 95.4%. After the deprotection and amination reaction, the yield of G-thamine was 6.5 wt.%. Finally, 6.4 wt.% (341 $\mu\text{mol/g}$) of dopamine hydrochloride was obtained based on spruce lignin.

Revised Fig. 5a The yields of the products in the spruce lignin-to-dopamine hydrochloride route. Mass yield = (mass of product) / (mass of lignin) \times 100%; Molar yield = (molar of product) / (mass of lignin); Yield of each step = (molar of product) / (molar of the substrate in each step) \times 100%; Depolymerization efficiency = (yield of G-acetal / theoretical yield of G-acetal) \times 100%; theoretical yield of G-acetal in lignin was obtained by combining the 2D-HSQC NMR method and derivatization followed by the reductive cleavage method.

In the tenfold scaling-up experiment with sole spruce lignin as substrate, the conversion of dopamine hydrobromide into dopamine hydrochloride was also carried out (**Revised Supplementary Fig. 21**). The obtained dopamine hydrochloride from spruce lignin was mixed with ethanol and heated at 80 $^{\circ}\text{C}$ in an oil bath under an N_2 atmosphere until fully dissolved. Then, HCl was slowly dripped into the solution and the reactor was cooled in an ice-water bath. Finally, the dopamine hydrobromide was obtained simply by filtration and drying. From HPLC results, the mass yield of dopamine hydrobromide was 3.3 wt.% based on lignin. In addition, the purity of dopamine hydrobromide from the tandem conversions of lignin was 94.7%. The impurities in products were also qualitatively and quantitatively analyzed, including 4-(2-aminoethyl) phenol hydrochloride (3.1%), 4-(2-aminoethyl)-2-methoxyphenol hydrochloride (1.1%), and dopamine hydrobromide (0.9%). No dimers or oligomers were detected, which were removed in the previous step *via* the column chromatography method.

The relevant discussion has now been updated in the revised Supplementary Information (page 8, lines 17-26).

Revised Supplementary Figure 21. Photos and process yield for the production of dopamine hydrochloride from spruce lignin as the sole substrate.

The conversion of dopamine hydrobromide into dopamine hydrochloride was also added to the process model to perform a techno-economic analysis (Revised Figs. 5b and 5c). The production cost of dopamine hydrochloride in this work was 2.20 million CNY/t (Chinese Yuan per ton) (Revised Supplementary Tables 6-9), which was much lower than the dopamine hydrochloride market prices in the range of 4-6 million CNY/t over the last year.

Revised Fig. 5b Conceptual process modeling for dopamine hydrochloride production from lignin. The process model integrates the four catalytic steps: (i) acid-catalyzed lignin depolymerization; (ii) deprotection reaction of G-acetal; (iii) hydrogen-borrowing amination of G-ethanol; and (iv) hydrolysis reaction of G-ethamine into dopamine hydrochloride.

Revised Fig. 5c Conceptual diagram for dopamine hydrochloride production from lignin. It is comprised of the production and application of dopamine hydrochloride.

Revised Supplementary Table 6. Assumptions for equipment cost.		
Item description	Equipment cost (CNY)	Equipment cost and other direct costs (CNY)
pump 1	395200	707200
Reactor 1	11719900	14831600
Cooler 1	53700	649100
Makeup H ₂ /N ₂	5565400	7017100
Pump 2	257300	418100
Reactor 2	11913100	15037300
Tower 1	1134900	3154000
Makeup NH ₃	2167600	2830500
Pump 3	271300	446500
Reactor 3	599800	1494063
Tower 2	1134900	3154000
Pump 4	255900	401200
Reactor 4	749400	1904600
Separation 1	1046900	1722700
Pump 5	255500	395800
Reactor 5	1001600	2194900
Separation 2	1046900	1722700
total	39569300	58081363

Revised Supplementary Table 7. Assumptions for electricity cost of equipment.			
Item description	Power (KW)	Electricity (KW·h)	Total cost (CNY)
pump 1	45	360000	180000
Reactor 1	150	1200000	600000
Cooler 1	3	24000	12000
Makeup H ₂ /N ₂	190	1520000	760000
Pump 2	3	24000	12000
Reactor 2	160	1280000	640000
Tower 1	--	--	--
Makeup NH ₃	1.11	8880	4440
Pump 3	7.5	60000	30000
Reactor 3	--	--	--
Tower 2	--	--	--
Pump 4	1.5	12000	6000
Reactor 4	22.7	181600	90800
Separation 1	15	120000	60000
Pump 5	1.5	12000	6000
Reactor 5	18.5	148000	74000
Separation 2	15	120000	60000
Total		5070480	2535240

Revised Supplementary Table 8. Assumptions for raw material cost.			
Items	Unit price (CNY/t)	Quantity(t/year)	Total cost (CNY/year)
Lignin	1000	10000	10000000
Sulfuric acid	514.9851	1464.4	754144.2
Ethylene glycol	8123.2934	234.9	1908161.6
1,4-dioxane	14900	15703	233974700
process water	15	50000	750000
N ₂	795.6	3500	2784600
H ₂	10000	500	5000000
Ru/C (5 wt.%)	3260400	250	815100000
Paraxylene	6902.2281	1546.3	10672915.3
NH ₃	2438.8256	75.9	185082.5
Ni/TiO ₂ (10wt%)	121857	1250	152321250
HBr	16000	59.13	946080
Ethanol	5000	55	275000
HCl	5000	125	625000
Total			1235296934

Revised Supplementary Table 9. Assumptions for the estimation of the total product cost			
Component		Base	Cost (CNY)
Equipment cost and other direct costs	1.1	Installed cost of all equipment	58081363
Indirect costs	1.2	60% of 1.1	34848817.8
Raw materials	2.1	All costs of raw materials and catalysts	1235296934
Utilities	2.2	Cooling water	3 CNY/t 820800
		Electricity	0.5 CNY/kW·h 2535240
Operating and maintenance cost	2.3.1	Operating labors	9 operators, 100000 CNY/operator/year 900000
	2.3.2	Direct supervisory and clerical labor	20% of (2.3.1) 180000
	2.3.3	Maintenance and repairs	10% of fixed capital 9293018.1
	2.3.4	Operating supplies	2% of fixed capital 1858603.6
	2.3.5	Laboratory charge	15% of (2.3.1) 135000
Depreciation	2.4	Straight line depreciation, life period 20y, salvage value 4%	4460648.7
Plant overhead cost	2.5	60% of 2.3.1+2.3.2+2.3.3	6223810.9
Administrative cost	2.6	2% of the product cost	27092684.7
Distribution and selling cost	2.7	2% of the product cost	27092684.7
Unit price of dopamine (CNY/t)			2201280.6

Q3. The purity is an important parameter of the product and the pathway. The authors should not simply mention that the product is of “high purity”, instead, an accurate data of purity should be given. Also, can the authors quantitatively identify the impurities in the product (especially the product from the tandem conversions of lignin)?

Response: Thanks very much for your valuable comment. **We have carried out a series of characterizations (e.g. HPLC, NMR, IC, and titrimetric analysis) to analyze the purity and impurities of obtained dopamine hydrochloride**, which can be referred to the Response to Question 2.

In the conversion of dopamine hydrobromide, the purity of obtained dopamine hydrochloride was 98.0%. The major impurities were 4-(2-aminoethyl)-2-methoxyphenol hydrochloride (0.9%) and dopamine hydrobromide (0.8%).

In the tandem conversion of spruce lignin, the purity of dopamine hydrobromide was 94.7%. The impurities in products were 4-(2-aminoethyl) phenol hydrochloride (3.1%), 4-(2-aminoethyl)-2-methoxyphenol hydrochloride (1.1%), and dopamine hydrobromide (0.9%). No

dimers or oligomers were detected, which were removed in the previous step *via* the column chromatography method.

The relevant discussion has now been updated in the revised manuscript (page 2, line 9; page 9, lines 16-21; page 10, lines 2 and 13) and Supplementary Information (page 8, lines 22-26).

Q4. Some details of the conversion procedure are still missing. For example, characterization techniques, such as TEM, XRD, IC... details in tenfold scaling-up experiments, such as product separation conditions.

Response: Thanks for your valuable suggestion. **As suggested, we have added the details of the conversion procedure to the revised Supplementary Information, including characterization techniques and tenfold scaling-up experiments.**

Transmission electron microscopy (TEM) was examined by using a JEOL JEM-2100EX microscope, and the electron-beam accelerating voltage was 200 kV. The powder X-ray diffraction (XRD) patterns were recorded on a Rigaku D/max-2550VB/PC diffractometer with Cu K α radiation ($\lambda = 1.5406 \text{ \AA}$). Each sample was scanned from $2\theta = 10^\circ$ to 80° . The analysis of Ni content in samples was performed by using inductively coupled plasma atomic emission spectrometry (ICP-AES, Agilent 725ES ICP-AES). The analysis of Cl and Br content in samples was performed by using ion chromatography (IC, DIONEX AQUION). IC analysis was performed using a Dionex IonPac AS19 column (4 \times 250 mm) with ampere detector. The mobile phase was a KOH aqueous solution (20 mmol/L) at a flow rate of 1.0 mL \cdot min $^{-1}$ and the column was kept at a temperature of 30 $^\circ$ C. Nitrogen adsorption/desorption isotherms were measured on a Micromeritics ASAP 2020M sorption analyzer. Before the measurements, all samples were outgassed at 180 $^\circ$ C for 12 h under vacuum to remove moisture and impurities from pores. The surface area was calculated by the Brunauer-Emmett-Teller (BET) method, the pore size distribution was calculated by the Barrett-Joyner-Halenda (BJH) method through the desorption branch of the isotherm and the total pore volume was estimated at a relative pressure of 0.975. Mass (MS) spectra were recorded using a SCIEX X500R LC-Q-TOF, ESI ion Source. Ultraviolet-visible (UV-Vis) spectra were recorded using a Perkin-Elmer Lambda 950. Fourier transform infrared (FT-IR) spectra were recorded on a Bruker VERTEX 80 V instrument. NMR spectra were recorded on Bruker Avance-400 instrument, using D $_2$ O as solvent. ^1H NMR spectral data were reported in terms of chemical shift (δ , ppm), multiplicity, coupling constant (Hz), and integration. ^{13}C NMR spectral data were reported in terms of chemical shift (δ , ppm). The following abbreviations indicated the multiplicities: s, singlet; d, doublet; t, triplet; q, quartet; m, multiplet; br, broad. The 2D-HSQC-NMR spectra were recorded on Bruker Avance-600 instrument. The test program follows: 10.3 ppm sweep width in the F2 (^1H), 165 ppm sweep

width F1 (^{13}C), an acquisition time of 130 ms, a relaxation delay time of 1.5 s, 24 scans and 512 data points. MestReNova software was used to process the HSQC data and quantitatively analyze the composition of the sample. **The relevant discussion has now been updated in the revised Supplementary Information (page 3, lines 23-30; page 4, lines 1-18).**

In tenfold scaling-up experiments (**Revised Supplementary Fig. 21**), we have added some details of the process of dopamine hydrochloride synthesis, in particular with product separation conditions. Firstly, the depolymerization of lignin into G-acetal was conducted in a 250 mL stainless-steel autoclave reactor. Spruce lignin (2.0 g) was mixed with ethylene glycol (7.2 mL), H_2SO_4 (160 μL), 1,4-dioxane (300 mL) and reacted at 140 $^\circ\text{C}$ for 10 h. After the reaction, the reactor was quenched in an ice-water bath and the mixtures were quantitatively analyzed by GC-FID. The yield of G-acetal was 9.5 wt.% based on the lignin. In this step, the obtained G-acetal did not need to be separated. Next, Ru/C (1.0 g) and H_2O (10 mL) were filled into the autoclave, which was sealed and charged to an initial pressure of 1.0 MPa with H_2/N_2 (50 vol%). The autoclave was heated to 120 $^\circ\text{C}$ for 10 h. After the deprotection reaction, the mixtures were analyzed by GC-FID and the yield of G-ethanol was up to 6.3 wt.%. The mixtures were centrifuged to separate the liquid phase and Ru/C catalyst, and the latter was recycled for the deprotection reaction. Reduced pressure distillation of the liquid phase was carried out to achieve 1,4-dioxane recovery, which was used for lignin depolymerization again. **To the resulting mixture, dichloromethane (10 mL) and saturated Na_2CO_3 solution in H_2O (20 mL) were added. The aqueous layer was extracted with dichloromethane (20 mL \times 3), and the combined organic layers were dried over anhydrous Na_2SO_4 , concentrated in vacuo, and subjected to silica gel flash chromatography (petroleum ether:ethyl acetate = 2:1) to afford the desired G-ethanol.** Then, the amination of G-ethanol was tested in a fixed-bed reactor. The 10%Ni/TiO₂ catalyst packed into the tubular reactor was *in-situ* reduced at 400 $^\circ\text{C}$ for 4 h. A feed of G-ethanol in *p*-xylene was injected into the fixed-bed reactor with a 0.163 h⁻¹ of weight-hourly space velocity (WHSV). The reactor was heated to 160 $^\circ\text{C}$ with an NH_3 flow rate of 100 mL/min. After the hydrogen-borrowing amination reaction, obtained G-ethamine was up to 5.1 wt.%. Reduced pressure distillation of the liquid phase was carried out and the recycled *p*-xylene solvent was used for the amination reaction. The crude products were concentrated in vacuo and further purified by column chromatography (dichloromethane:methanol = 100:1) leading to highly purified G-ethamine. The G-ethamine was mixed with HBr and reacted at 120 $^\circ\text{C}$ for 6 h in an oil bath under an N_2 atmosphere. After the hydrolysis reaction, the mixtures were centrifuged to separate the liquid phase and solid insoluble precipitate, and the former was recycled for the hydrolysis reaction. The dopamine hydrobromide solid was obtained simply by drying at 50 $^\circ\text{C}$ in a vacuum oven. The obtained

dopamine hydrochloride was mixed with ethanol and heated at 80 °C in an oil bath under N₂ atmosphere until fully dissolved. Then, HCl was slowly dripped into the solution and the reactor was cooled in an ice-water bath. Finally, the dopamine hydrochloride solid was obtained simply by filtration and drying. The liquid phase was recycled for the conversion of dopamine hydrobromide. From HPLC results, the mass yield of dopamine hydrobromide was 3.3 wt.% based on spruce lignin. In addition, the purity of dopamine hydrobromide from the tandem conversions of lignin was 94.7%. The impurities in products were also qualitatively and quantitatively analyzed, including 4-(2-aminoethyl)phenol hydrochloride (3.1%), 4-(2-aminoethyl)-2-methoxyphenol hydrochloride (1.1%), and dopamine hydrobromide (0.9%). No dimers or oligomers were detected, which were removed in the previous step *via* column chromatography method. The relatively low yield of dopamine hydrochloride might be due to mass loss in the process of transfer, separation, and purification. We are still working on the improvement of the overall efficiency including product purification, solvent recovery, catalyst, and additive separation and recovery.

The relevant discussion has now been updated in the revised Supplementary Information (page 7, line 26; page 7, lines 32-35; page 8, lines 8-26).

Revised Supplementary Figure 21. Photos and process yield for the production of dopamine hydrochloride from spruce lignin as the sole substrate.

Q5. The production cost is calculated to be 2.23 million CHY/t, which is still very high to me. The author can give a brief expectation on reducing the cost.

Response: Thank you for this critical suggestion. The production cost of dopamine hydrochloride in this work was recalculated to be 2.20 million CNY/t (Chinese Yuan per ton), which was much lower than the dopamine hydrochloride market prices in the range

of 4-6 million CNY/t over the last year (available at the link of <https://www.cognitivemarketresearch.com/dopamine-market-report>). Based on the results of the lignin-to-dopamine hydrochloride route and techno-economic analysis, we highlighted the following aspects to further reduce the cost for future consideration.

a) It is crucial to take into consideration of the costs of solvents, especially for large-scale applications in order to make the biorefineries be feasible economically. The substrate concentration was not high in this work, which require additional cost of solvents and energy for product separation. Thus, reaction process conditions should be further optimized to increase the concentration of substrates.

b) Due to the use of homogeneous acids (*e.g.* H₂SO₄, HBr, and HCl), the used equipment must have excellent corrosion resistance. Developing catalytic systems with heterogeneous solid acid catalysts instead of homogeneous acids is highly desired to reduce equipment costs. In addition, the heterogeneous solid acid catalysts are easy to be separated and recycled compared to the homogeneous ones.

c) In our processes, acid-catalyzed lignin depolymerization proceeds via cleavage of the β -O-4 bond to afford low-molecular-weight monomers, but the other interunit C-O linkages (*e.g.* 4-O-5 and α -O-4) and C-C linkages are not cleaved. Because of the presence of stable interunit C-O and C-C bonds within native lignin, the yield of dopamine hydrochloride from the tandem conversions of lignin is seriously limited. Designing innovative and new strategies that can conduct cleavage of both interunit C-O and C-C linkages is believed to mitigate the limitations on dopamine hydrochloride production.

d) After lignin depolymerization, the desired G-acetal was about 10 wt.% based on lignin and the remaining by-products are mainly phenolic oligomers. The residual phenolic oligomers can be further converted into highly valuable chemicals to achieve a high mass and carbon efficiency, such as printing ink, fuels, polymeric materials, and so on (*Science* **367**, 1385-1390 (2020); *Appl. Catal. B: Environ.* **263**, 118325 (2020); *Chem. Rev.* **116**, 2275-2306 (2016)). Maximal valorization of lignin into value products is key in demonstrating the potential of wood biorefineries.

The relevant discussion has now been updated in the revised Supplementary Information (page 8, lines 30-35; page 9, lines 1-19).

Q6. The words “Step 3” is missing in Fig. 5b.

Response: We apologize for the mistake. **We have corrected this in the revised Fig. 5b.**

Q7. Citations should be given for Fig. S1 and S2.

Response: Thanks for your suggestion. **We have added citations for Supplementary Fig. 1**

and Supplementary Fig. 2 in the revised Supplementary Information (page 10).

Revised Supplementary Figure 1. The diverse application pattern of dopamine in pharmaceutical and material synthesis. The citations in the figure are listed in References.

Revised Supplementary Figure 2. Schematic representation of the conversion of lignin towards bio-based amines based on previous literature. The citations in the figure are listed in References.

References

1. Franco, R., Reyes-Resina, I. & Navarro, G. Dopamine in health and disease: much more than a neurotransmitter. *Biomedicines* **9**, 109 (2021).
2. Berke, J. D. What does dopamine mean? *Nat. Neurosci.* **21**, 787-793 (2018).
3. Piggott, M. A. *et al.* Striatal dopaminergic markers in dementia with Lewy bodies, Alzheimer's and Parkinson's diseases: rostrocaudal distribution. *Brain* **122**, 1449-1468 (1999).
4. Johnson, K. A. *et al.* Combined dopamine transporter and FDG PET in dementia with Lewy bodies, Alzheimer's disease, and Parkinson's disease. *Neurology* **62**, A300-A300 (2004).
5. Zhang, A., Neumeier, J. L. & Baldessarini, R. J. Recent progress in development of dopamine receptor subtype-selective agents: potential therapeutics for neurological and psychiatric disorders. *Chem. Rev.* **107**, 274-302 (2007).
6. Lee, H. A., Ma, Y., Zhou, F., Hong, S. & Lee, H. Material-independent surface chemistry beyond polydopamine coating. *Acc. Chem. Res.* **52**, 704-713 (2019).
7. Mei, S., Xu, X., Priestley, R. D. & Lu, Y. Polydopamine-based nanoreactors: synthesis and applications in bioscience and energy materials. *Chem. Sci.* **11**, 12269-12281 (2020).
8. Natte, K., Narani, A., Goyal, V., Sarki, N. & Jagadeesh, R. V. Synthesis of functional chemicals from lignin-derived monomers by selective organic transformations. *Adv. Synth. Catal.* **362**, 5143-5169 (2020).
9. Chen, Z. W., Zeng, H. Y., Gong, H., Wang, H. N. & Li, C-J. Palladium-catalyzed reductive coupling of phenols with anilines and amines: efficient conversion of phenolic lignin model monomers and analogues to cyclohexylamines. *Chem. Sci.* **6**, 4174-4178 (2015).
10. Pelckmans, M., Renders, T., Van de Vyver, S. & Sels, B. F. Bio-based amines through sustainable heterogeneous catalysis. *Green Chem.* **19**, 5303-5331 (2017).
11. Sun, Z. H. *et al.* Complete lignocellulose conversion with integrated catalyst recycling yielding valuable aromatics and fuels. *Nat. Catal.* **1**, 82-92 (2018).

Reviewers' Comments:

Reviewer #3:

Remarks to the Author:

The comments have been well addressed. I'm glad to recommend acceptance for publication.